# Learn Globally, Speak Locally: Bridging the Gaps in Multilingual Reasoning

## Abstract

Large Language Models (LLMs) have achieved strong performance in domains like mathematics, factual question answering, and code generation, yet their ability to reason on these tasks in different languages remains underdeveloped. Especially for low-resource languages such as Swahili or Thai, LLMs can often misinterpret prompts or default to reasoning in English. This implicit bias toward high-resource languages undermines factual accuracy, interpretability, and trust. We propose M2A, a novel method that combines multi-scale multilingual alignment with language-consistency rewards on machine-translated questions, training models to reason directly and accurately in the target language. Furthermore, existing multilingual benchmarks only evaluate on final answers, overlooking whether reasoning occurs in the intended language. To close this gap, we introduce GEOFACT-X, a geography-based multilingual factual reasoning benchmark together with reasoning traces in five languages: English, Hindi, Japanese, Swahili, and Thai. Our results show that M2A significantly enhances multilingual reasoning fidelity in both mathematical and factual reasoning tasks, highlighting that reasoning-aware multilingual reinforcement learning is crucial for robust cross-lingual generalization.

## 1 Introduction

Large Language Models (LLMs) have made remarkable progress in reasoning tasks, such as mathematics (Liu et al., 2024; Shao et al., 2024), code generation (Jain et al., 2024; Team et al., 2025), and factual QA (Achiam et al., 2023; Guo et al., 2025; Qwen et al., 2025; Wang et al., 2024), primarily in English. Yet, their reasoning capabilities remain underdeveloped in low-resource languages such as Swahili, Marathi, or Thai (Cahyawijaya et al., 2024; Nguyen et al., 2023). This performance disparity undermines the *trustworthiness* of LLMs in these languages since users cannot check the reasoning traces to verify the answer. For this purpose, both the final answer and the intermediate reasoning should ideally be expressed in the question language to ensure *interpretability*, *i.e.*, users can directly follow the reasoning in their own language. The central issue is not only whether LLMs can provide correct answers, but whether they can *think* in the language of the question. When they cannot, translation of reasoning traces offers only a partial solution and one that often fails on cultural and linguistic nuance. Recent studies (Aggarwal et al., 2025; Yao et al., 2024) suggest that both LLMs and machine translation systems struggle with cultural and linguistic nuances. For example, culturally grounded concepts such as Chinese *Guānxi* (系), Japanese *Wa* (和), and Korean *Jeong* (정) remain difficult to capture faithfully.

In this work, we conduct the first comprehensive study of **multilingual reasoning**: assessing whether LLMs can not only answer questions correctly, but also *reason in the same language as the question*. Prior multilingual benchmarks primarily assess the final accuracy (Ponti et al., 2020; Shi et al., 2023; Xuan et al., 2025), overlooking the language of the reasoning traces. By evaluating reasoning traces directly on MGSM (Shi et al., 2023), we found that models often revert to English reasoning even under non-English prompts. This gap between the prompt language and the reasoning process underscores a broader problem that LLMs can appear correct while failing to reason in a language. Ensuring both accuracy and language-consistent reasoning is essential for globally inclusive and interpretable AI.

To tackle this challenge, we introduce M2A (Multi-Scale Multilingual Alignment), an efficient approach that explicitly enforces language-consistent reasoning while retaining factual correctness.

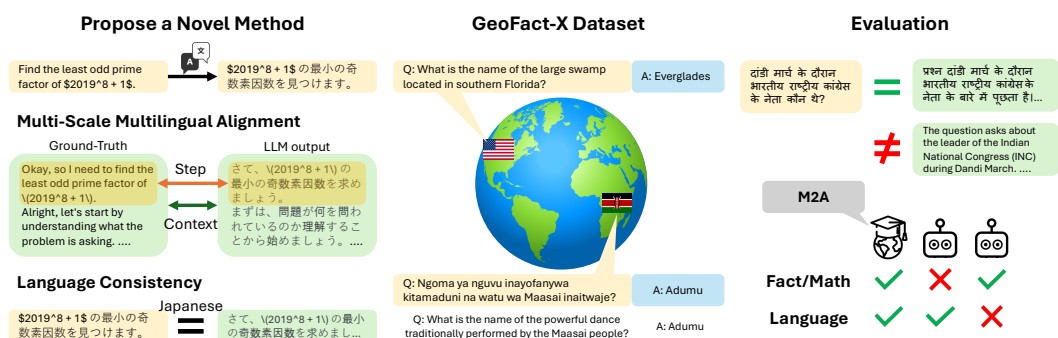

Figure 1: **Illustration of Contributions** We propose M2A, a new method that utilizes multi-scale multilingual alignment and language consistency rewards from a given machine-translated question, enabling reasoning in the question language. We also introduce GEOFACT-X, a new multilingual factual reasoning benchmark which includes training datasets and step-by-step reasoning traces across five languages. We propose an automatic evaluation protocol to assess whether a model reasons in the question language and the correctness of reasoning via language identifier or LLM-as-a-judge.

The key idea is to combine multi-scale multilingual reasoning alignment with a language-consistency reward, providing reinforcement-learning signals that encourage reasoning traces to remain in the question language, enabling reasoning capabilities to be learned without ground-truth supervision in that language. We jointly employ supervised fine-tuning and group relative policy optimization (GRPO) to integrate supervised learning on ground-truth reasoning traces with reinforcement-based refinement. Unlike prior work (Guo et al., 2025; Liu et al., 2025; Ranaldi & Pucci, 2025) that optimizes only for correctness or formality, our method targets the *alignment of reasoning itself*.

Despite recent progress in multilingual LLMs (Ahuja et al., 2023; Qin et al., 2025), their ability to perform factual reasoning across cultural contexts remains largely unevaluated. We introduce GEOFACT-X, a benchmark of culturally grounded questions localized to five countries (USA, India, Japan, Kenya, Thailand) in their predominant languages (English, Hindi, Japanese, Swahili, Thai). By grounding evaluation in country-specific knowledge, GEOFACT-X enables systematic assessment of whether LLMs can reason faithfully within linguistically and culturally contextualized spaces.

We train M2A on the s1K-1.1 (Muennighoff et al., 2023) dataset and GEOFACT-X train set for mathematical and factual reasoning, respectively. Our experiments demonstrate that M2A yields significant improvement on multilingual reasoning in both cases while reasoning in the question languages. Figure 1 summarizes our key contributions. Together with the release of our code, data, and evaluation protocols, our work provides a foundation for future work on multilingual reasoning.

## 2 RELATED WORK

### 2.1 MULTILINGUALISM IN LARGE LANGUAGE MODELS

Recent advances in multilingual large language models (LLMs) mark a shift from monolingual dominance to more inclusive cross-lingual capabilities. Two perspectives frame much of this discourse. The scaling view holds that increasing the volume and diversity of multilingual data during pretraining enhances cross-lingual generalization (Xue et al., 2021; Conneau et al., 2020; Chang et al., 2023; Gurgurov et al., 2024), though it faces the *curse of multilinguality*, where accommodating many languages dilutes performance in individual ones due to limited parameter capacity. The optimization view emphasizes careful fine-tuning as a means of preserving and amplifying multilingual knowledge (Devlin et al., 2019; Conneau et al., 2020; Luo et al., 2023; Zhai et al., 2023), yet aggressive post-training risks catastrophic forgetting by overwriting deeply embedded linguistic priors. Beyond scaling and optimization, Schut et al. (2025) and Zhong et al. (2025) analyzed the internal representations of multilingual models, while Yong et al. (2025) proposed test-time strategies for improving cross-lingual reasoning. Notably, Schut et al. (2025) found that multilingual LLMs often perform intermediate reasoning in English, but their scope was limited to internal representation analysis rather than generated reasoning traces. More recently, post-training alignment techniques

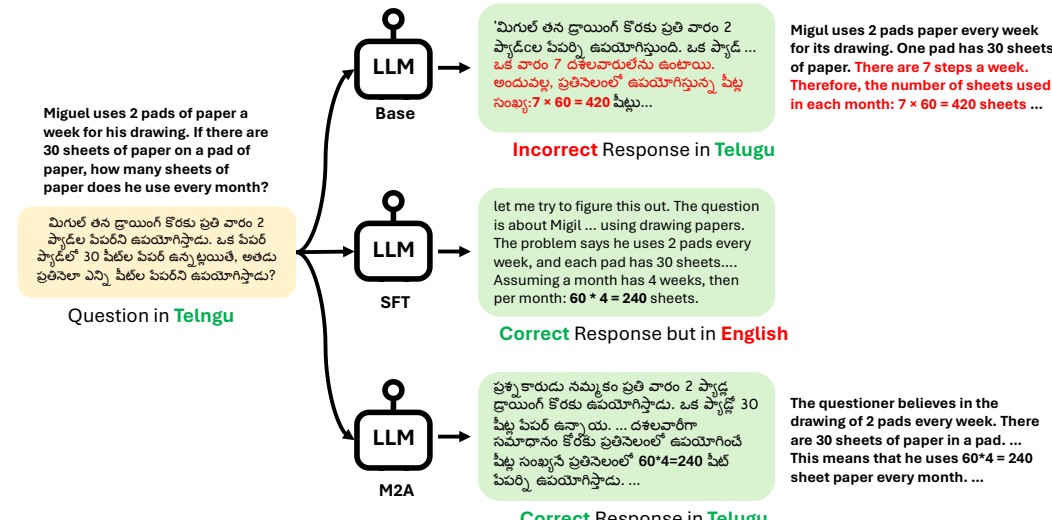

Figure 2: **Example of outputs from a Telugu question from three different models.** Base LLM and supervised fine-tuned models (SFT) are correct only in either answer or language, whereas our M2A is correct in both answer and language.

such as Direct Preference Optimization (DPO) (Rafailov et al., 2023) have been applied to embed multilingual reasoning (Dang et al., 2024; Ranaldi & Pucci, 2025). Similarly, we employ Group Relative Policy Optimization (GRPO) (Guo et al., 2025; Shao et al., 2024) with multi-scale multilingual alignment and language consistency rewards, leveraging explicit reasoning traces in the original languages to build stronger multilingual reasoning capabilities.

## 2.2 EVALUATION OF MULTILINGUAL REASONING CAPABILITIES

Instruction-tuning datasets such as Bactrian (Li et al., 2023), Aya (Singh et al., 2024), Multilingual Alpaca (Chen et al., 2023), and SphinX (Ahuja et al., 2024) have improved performance across high- and low-resource languages by emphasizing diversity and cultural specificity. Complementary to these efforts, benchmarks like MEGA (Ahuja et al., 2023) provide broad multilingual coverage across 70 languages and 16 NLP tasks, but primarily evaluate task-level accuracy rather than reasoning processes. Other multilingual reasoning benchmarks, including XCOPA (Ponti et al., 2020), XWinograd (Tikhonov & Ryabinin, 2021), and XStoryCloze (Lin et al., 2022), adopt multiple-choice formats that permit shallow guessing and suffer from translation artifacts (Li et al., 2024). In contrast, our benchmark directly targets multilingual reasoning by evaluating free-form, step-by-step generation with explicit reasoning traces. This design enables more faithful assessment of both reasoning quality and language alignment, providing a sharper diagnostic tool for multilingual LLMs.

## 3 M2A: MULTI-SCALE MULTILINGUAL ALIGNMENT

We propose a new method to enrich existing English-based reasoning models with multilingual reasoning capabilities. Our approach combines the complementary strengths of supervised fine-tuning (SFT) and reinforcement learning (RL). While SFT enables base reasoning capabilities in English, we find that these models struggle with multilingual reasoning during test time as shown in Figure 2. To that end, we propose a new test-time RL method for multilingual reasoning. Key to this new approach is defining the right set of rewards that incentivize the model to reason consistently across different languages. We first translate each question into multiple languages by using Google Translate, allowing the model to generate outputs conditioned on the translated inputs. We then define a set of multi-scale rewards across different multilingual granularities. A context alignment reward measures multilingual alignment across the entire reasoning context with the original ground-truth reasoning trace. This is followed by a reasoning-step alignment reward that aligns each individual reasoning step to capture fine-grained correspondence. Finally, a language consistency reward explicitly enforces reasoning in the question language.

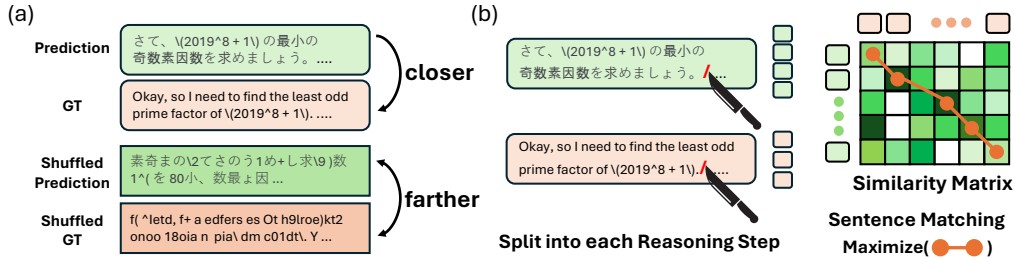

Figure 3: **Overview of M2A.** (a) *Multilingual Context Alignment* enforces global similarity between generated and reference responses while discouraging trivial matches via shuffled negatives. (b) *Multilingual Reasoning-Step Alignment* provides finer-grained supervision by aligning individual reasoning steps with ground-truth traces using dynamic programming.

**Problem Setup.** Given the question sampled from the question dataset, $q \sim P(\mathcal{Q})$ and its corresponding ground-truth response $y$, we translate each question to the target language $l$, $q'$ by using a machine translator (*e.g.*, Google Translate). GRPO generates a group of outputs $\{o_1, o_2, ..., o_G\}$ from the translated question. The reward $r_t$ is calculated based on each output $o_t$.

**Multilingual Context Alignment.** We first encode both output and the ground-truth with the encoder, $\phi$, $z_o = \phi(o)$ and $z_y = \phi(y)$. We utilize mT5 (Xue et al., 2021) for encoding texts. The alignment reward can be the cosine similarity between two embeddings, $\cos(z_o, z_y)$. However, it is maximized when the generated output $o$ is identical to $y$, ignoring the question language. To address this, we introduce negative samples by shuffling both outputs and ground-truth responses with the same permutation, $\tilde{z}_o = \phi(\psi(o))$ and $\tilde{z}_y = \phi(\psi(y))$, where $\psi$ denotes the shuffle function. Inspired by Schroff et al. (2015), the multilingual context alignment maximizes similarity between positive samples and minimizes similarity between negative samples, enforcing a margin, $\alpha$, between these similarities:

$$\cos(\tilde{z}_o, \tilde{z}_y) + \alpha < \cos(z_o, z_y). \tag{1}$$

The final context alignment reward is defined as:

$$r_{\text{context-align}} = \max(\cos(z_o, z_y) - \cos(\tilde{z}_o, \tilde{z}_y) + \alpha, 0), \tag{2}$$

where $\alpha$ denotes the margin, set to 1, the maximum possible value of cosine similarity.

**Multilingual Reasoning-Step Alignment.** We further introduce a multilingual reasoning-step alignment to provide finer-grained matching. Given the split output sentences, $\mathbf{o} = (o^{(1)}, \ldots, o^{(N)})$ and ground-truth sentences $\mathbf{y} = (y^{(1)}, \ldots, y^{(M)})$, each output sentence, $o^{(i)}$ is aligned with a ground-truth sentence, $y^{(j_i)}$. Since the number of output and reference sentences ($N$ and $M$) may differ, we use dynamic programming to maximize the total similarity between the pairs while preserving order:

$$\max_{1 \leq j_1 \leq \ldots \leq j_N \leq M} \sum_{i=1}^{N} \mathbf{C}_{i,j_i}, \tag{3}$$

where $\mathbf{C} \in \mathbb{R}^{N \times M}$ is the similarity matrix, and $\mathbf{C}_{i,j}$ denotes the similarity score between embeddings $z_o^{(i)}$ and $z_y^{(j)}$. We use the same function used in Eq. (2) for $\mathbf{C}$. The multilingual reasoning-step alignment reward is then defined as the average similarity across aligned pairs:

$$r_{\text{step-algin}} = \frac{1}{N} \sum_{i=1}^{N} \mathbf{C}_{i,j_i} = \frac{1}{N} \sum_{i=1}^{N} \max(\cos(z_o^{(i)}, z_y^{(j_i)}) - \cos(\tilde{z}_o^{(i)}, \tilde{z}_y^{(j_i)}) + \alpha, 0). \tag{4}$$

**Language Consistency.** We also define a language consistency reward for giving a more direct incentive to reason in the question language. Given the output $o$, and language detector $f$ (*e.g.*, Google Translate, langid (Lui & Baldwin, 2012)), the language consistency reward is defined as $1$ if the detected language in response matches the target language $l$, $0$ if it does not:

$$r_{\text{lang}} = \delta[f(o) = l_t], \tag{5}$$

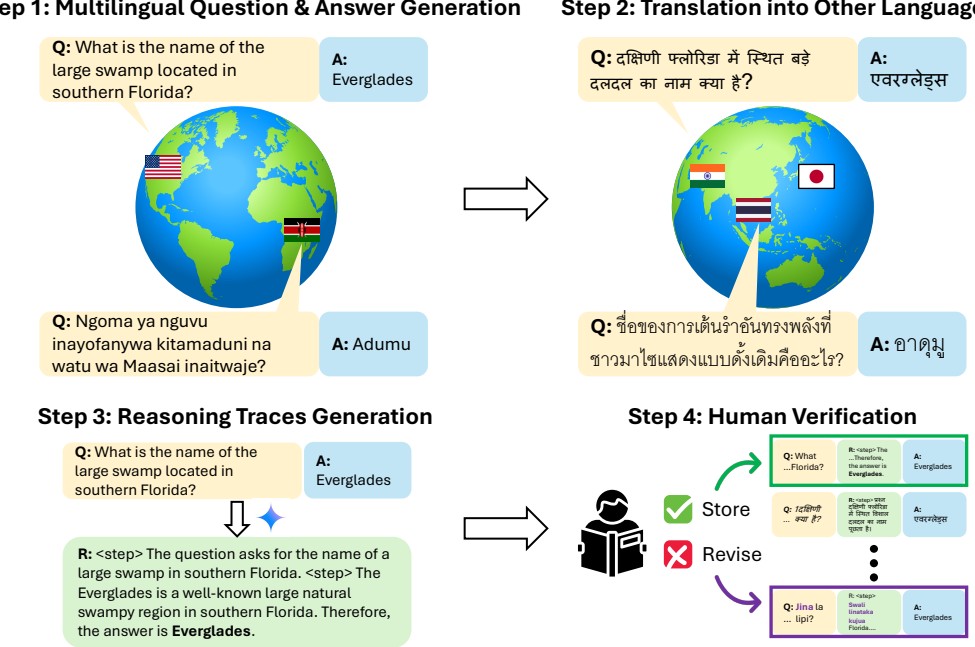

Figure 4: **Illustration of GEOFACT-X benchmark construction.** (1) Geography-aware multilingual questions and answers are generated by Gemini 2.0 Flash. (2) The data is translated into other languages, verifying whether it is back-translatable. (3) The reasoning trace for each question and answer pair is generated. (4) Native or C1-level speakers verify each data and revise it if needed.

Table 1: Comparison between existing multilingual factual or common-sense benchmark and GEOFACT-X.

| Benchmark | Size | #Lang. | Geo-Aware | Train Set | Reasoning Eval. |
|---|---|---|---|---|---|
| XStoryCloze (Lin et al., 2022) | 1872 | 11 | | ✓ | |
| XWINO (Tikhonov & Ryabinin, 2021) | 3961 | 6 | | | |
| XCOPA(Ponti et al., 2020) | 6600 | 11 | | ✓(English only) | |
| X-FaKT(Aggarwal et al., 2025) | 2362 | 13 | | | |
| XLQA (Roh et al., 2025) | 3000 | 8 | ✓ | | |
| GEOFACT-X (ours) | 12780 | 5 | ✓ | ✓ | ✓ |

where $\delta[\cdot]$ denotes the indicator function. The final reward is defined as the sum of individual reward:

$$r = r_{\text{context-align}} + r_{\text{step-align}} + r_{\text{lang}}. \tag{6}$$

## 4 GEOFACT-X: GEOGRAPHY-BASED FACTUAL REASONING BENCHMARK

Despite advances in multilingual LLMs (Ahuja et al., 2023; Qin et al., 2025), robust evaluation of factual reasoning across cultures remains underexplored. We introduce GEOFACT-X, a benchmark of 3,000 culturally grounded questions (about 600 per country) spanning history, politics, geography, art, and culture, localized to the USA, India, Japan, Kenya, and Thailand in their predominant languages (English, Hindi, Japanese, Swahili, and Thai). Our goal is to capture country-specific factual knowledge, encouraging language models to reason effectively within culturally contextualized knowledge spaces. Table 1 compares GEOFACT-X with existing multilingual factual reasoning benchmarks. Our geography-aware multilingual benchmark has a training set and reasoning evaluations compared to other benchmarks.

Figure 4 illustrates the process of the dataset construction. We adopt a two-stage validation pipeline to ensure factual accuracy and dataset quality. Rule-based filters and cross-language checks remove incorrect or inconsistent pairs. Specifically, we verify cross-language answer consistency by translating each answer into English via the Google Translate API to identify mismatches. Gemini 2.0

Flash (Team et al., 2023) then generates structured chain-of-thought reasoning traces for each item, enhancing interpretability and providing supervision signals. We split the dataset into a train set (85%) and a test set (15%), ensuring no semantic overlap across splits, even across languages. All test samples are manually verified by native or C1-level speakers for factual correctness and linguistic clarity. Figure 6 illustrates an example multilingual question with its reasoning trace and final answer.

For evaluation, we measure answer accuracy and reasoning score. Answer accuracy is computed by comparing predictions against the reference answers. The reasoning score is assessed by Qwen-2.5-72B-Instruct (Qwen et al., 2025) as an LLM-as-a-judge, comparing model-generated reasoning traces against the human-verified reasoning traces in the test set. If a reasoning trace is produced in a language different from the question, identified by a language detector (Lui & Baldwin, 2012), its score is set to zero. We validate the reliability of this metric via a human agreement study in Appendix H. Detailed curation, distribution, and evaluation procedures are provided in Appendix A.

## 5 Revisiting Multilingual Mathematical Reasoning Benchmark

We investigate whether strong performance on multilingual reasoning benchmarks reliably reflects reasoning in the question language. As a case study, we use MGSM (Shi et al., 2023), which evaluates multilingual mathematical reasoning in ten diverse languages and provides chain-of-thought prompts (Naive-CoT) in each language to enforce reasoning in the language. MGSM reports only mathematical accuracy, implicitly assuming that high accuracy implies language-consistent reasoning.

To address this, we introduce *language accuracy*, which measures whether the generated reasoning matches the intended question language. Formally, given the language identifier (*e.g.*, Google Translate, langid (Lui & Baldwin, 2012)), $f$, language accuracy, $A_{\text{lang}}$ is defined as follows:

$$A_{\text{lang}} = \frac{1}{N} \sum_{n}^{N} \delta[f(o_n) = l_n], \tag{7}$$

where $N$ denotes the number of samples in the dataset, and $\delta[\cdot]$ is indicator function. $o_n$, and $l_n$ mean the generated output and the target question language, respectively. Then, we defined the joint accuracy of mathematics and language, $A_{\text{joint}}$ as follows:

$$A_{\text{joint}} = \frac{1}{N} \sum_{n}^{N} \left( \delta[f(o_n) = l_n] \cdot \delta[\hat{a}_n = a_n] \right), \tag{8}$$

where $\hat{a}_n$ and $a_n$ indicate predicted and ground-truth answers for $n$-th sample, respectively.

We evaluate various recent large language models, including Qwen2.5 (Hui et al., 2024), Llama3 (Grattafiori et al., 2024), Gemma3 (Team et al., 2025), and DeepSeek-R1 (Guo et al., 2025), on MGSM (see Appendix B.3 for the full list). Figure 5 illustrates average mathematical accuracy against joint accuracy across different languages. Ideally, both metrics should be the same (grey dashed line), yet models such as Qwen2.5-72B-Math-Instruct and Llama-3-70B-Instruct show large gaps, indicating frequent reasoning in the wrong language. Moreover, the s1 models (orange), fine-tuned from Qwen2.5-Instruct (green), notably degrade language accuracy while improving mathematical performance. These results demonstrate that mathematical accuracy alone overestimates multilingual reasoning ability, and joint evaluation is essential for assessing true language-consistent reasoning.

## 6 Experiments

We use Qwen-2.5-7B-Instruct as the backbone for all experiments on mathematical and factual reasoning. Training and evaluation are conducted on 4 NVIDIA A100 GPUs with DeepSpeed (Rasley et al., 2020). We use three random seeds to calculate the mean and standard error. Please refer to Appendix B and the attached codebase for the implementation and training details.

### 6.1 Dataset

**Mathematical Reasoning.** The s1K-1.1 dataset (Muennighoff et al., 2025) contains 1,000 curated math questions with chain-of-thought traces, selected for difficulty, diversity, and quality. To test

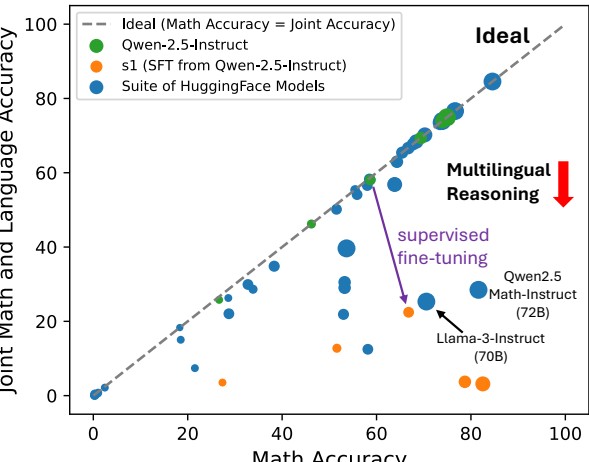

Figure 5: **Mathematical accuracy and the joint accuracy of mathematics and language of various LLMs on MGSM with native Chain-of-Thought.** Circle size is proportional to the number of parameters. The $y = x$ line represents ideal performance, where a model always uses the target question language in reasoning. Many models, especially the supervised fine-tuned model, s1, fall significantly below this line, indicating they solve the problem correctly but fail to adhere to the language prompt.

Table 2: Accuracy of Qwen2.5-7B-Instruct and post-training methods in GSM8K (English) and MGSM (ten languages). Results are reported for mathematical accuracy (Math.), language accuracy (Lang.), and joint accuracy (Joint). Bold indicates the best performance in each column.

| Method | GSM8K | | | MGSM | | |
|---|---|---|---|---|---|---|
| | Math. | Lang. | Joint | Math. | Lang. | Joint |
| Qwen-2.5-Instruct | 81.2 | **100** | 81.2 | 58.7 | 99.0 | 58.1 |
| GRPO | 80.4 ± 0.9 | **100.0 ± 0.0** | 80.4 ± 0.9 | 58.8 ± 0.4 | 95.9 ± 2.9 | **58.2 ± 0.7** |
| SFT (s1) | 87.2 ± 1.6 | **100.0 ± 0.0** | 87.2 ± 1.6 | **66.7 ± 0.1** | 31.0 ± 0.5 | 21.9 ± 0.6 |
| SFT on s1K-X | 84.3 ± 1.1 | 66.7 ± 33.3 | 56.5 ± 28.3 | 45.2 ± 4.1 | **99.7 ± 0.1** | 45.0 ± 4.3 |
| M2A (ours) | **87.3 ± 0.1** | **100.0 ± 0.0** | **87.3 ± 0.1** | 59.0 ± 0.3 | 97.8 ± 0.2 | 58.1 ± 0.4 |

multilingual generalization, we additionally construct S1K-X, a multilingual version of s1K-1.1 obtained by translating into ten typologically diverse languages via Google Translate, used for baseline SFT results. For evaluation, we report results on GSM8K (Cobbe et al., 2021) and its multilingual counterpart MGSM (Shi et al., 2023) with Native CoT prompts. We also evaluate language accuracy introduced in Section 5.

**Factual Reasoning.** We utilize GEOFACT-X (Section 4), which contains culturally grounded factual QA pairs across five countries (USA, India, Japan, Kenya, Thailand) in five local languages (English, Hindi, Japanese, Swahili, Thai). Models are trained on the train split and evaluated on the test split.

## 6.2 MATHEMATICAL REASONING

Table 2 presents the performance of the base model, Qwen2.5-7B-Instruct, and models with post-training methods, supervised fine-tuning (SFT), GRPO, and M2A. Supervised fine-tuning on s1K-1.1 improves mathematical reasoning performance on GSM8K and MGSM but substantially degrades multilingual performance in MGSM, leading to lower joint accuracy. Training on the translated multilingual dataset (s1K-X) preserves language accuracy on MGSM but reduces mathematical accuracy. GRPO, in contrast, produces little change, likely due to sparse rewards. For instance, Figure 11 shows that GRPO outputs are identical to the base model, whereas SFT produces an English response to a Russian query.

M2A outperforms baselines in all metrics on GSM8K and achieves large gains in joint accuracy on MGSM compared to SFT. Unlike SFT, it preserves reasoning in the query language while

Table 3: Comparison of model performance on average reasoning score (%), language accuracy (%), and answer accuracy (%) on GEOFACT-X test set, evaluated across all examples and split by whether the language is associated with the country ('Assoc.') or not ('Non-Assoc.'). Bold means the best performance.

| Model | Average Reasoning Score (%) | | | Average Answer Accuracy (%) | | |
|---|---|---|---|---|---|---|
| | All | Assoc. | Non-Assoc. | All | Assoc. | Non-Assoc. |
| DeepSeek-R1-Distill-Llama-8B | 13.8 | 16.7 | 13.1 | 8.4 | 10.8 | 7.8 |
| DeepSeek-R1-Distill-Qwen-7B | 13.8 | 16.7 | 13.1 | 7.2 | 10.3 | 6.4 |
| Command R7B | 33.1 | 40.3 | 31.2 | 25.8 | 33.7 | 23.8 |
| Qwen-2.5-Instruct | 30.4 | 38.5 | 28.3 | 26.2 | 33.7 | 24.3 |
| GRPO | $45.4 \pm 0.2$ | $48.1 \pm 0.1$ | $44.8 \pm 0.3$ | $32.1 \pm 0.3$ | $37.6 \pm 0.2$ | $\mathbf{30.7 \pm 0.3}$ |
| SFT | $47.6 \pm 0.1$ | $50.7 \pm 0.5$ | $46.9 \pm 0.1$ | $29.3 \pm 0.2$ | $37.1 \pm 1.0$ | $27.3 \pm 0.3$ |
| SFT + GRPO | $26.9 \pm 0.9$ | $29.2 \pm 0.9$ | $26.4 \pm 0.9$ | $10.7 \pm 0.9$ | $14.7 \pm 1.7$ | $9.7 \pm 0.7$ |
| M2A (ours) | $48.5 \pm 0.4$ | $52.6 \pm 0.5$ | $47.5 \pm 0.3$ | $32.0 \pm 0.6$ | $\mathbf{41.3 \pm 1.0}$ | $29.7 \pm 0.5$ |
| M2A (ours, Thai only) | $\mathbf{49.8 \pm 0.3}$ | $\mathbf{53.4 \pm 0.2}$ | $\mathbf{48.8 \pm 0.5}$ | $\mathbf{32.2 \pm 0.4}$ | $39.9 \pm 0.4$ | $30.2 \pm 0.3$ |

Table 4: Machine-translated performance of each model on GEOFACT-X test set. Google Translate is used to translate the generated output into the question language. Bold means the best performance.

| Model with Machine Translation | Average Reasoning Score (%) | | | Average Answer Accuracy (%) | | |
|---|---|---|---|---|---|---|
| | All | Assoc. | Non-Assoc. | All | Assoc. | Non-Assoc. |
| DeepSeek-R1-Distill-Llama-8B | 27.6 | 29.2 | 27.2 | 7.9 | 11.1 | 7.1 |
| DeepSeek-R1-Distill-Qwen-7B | 33.2 | 33.9 | 33.1 | 8.8 | 11.1 | 8.2 |
| Command R7B | 44.2 | 48.5 | 43.1 | 25.0 | 34.1 | 22.7 |
| Qwen-2.5-Insturct | 45.7 | 49.2 | 44.8 | 28.9 | 36.4 | 27.0 |
| GRPO | $45.7 \pm 0.3$ | $48.3 \pm 0.3$ | $45.1 \pm 0.3$ | $\mathbf{31.9 \pm 0.3}$ | $37.6 \pm 0.8$ | $\mathbf{30.5 \pm 0.2}$ |
| SFT | $47.8 \pm 0.1$ | $50.9 \pm 0.4$ | $47.1 \pm 0.1$ | $27.2 \pm 0.1$ | $35.8 \pm 1.1$ | $25.0 \pm 0.3$ |
| SFT + GRPO | $47.7 \pm 0.2$ | $51.1 \pm 0.2$ | $46.8 \pm 0.3$ | $34.0 \pm 0.1$ | $39.1 \pm 0.7$ | $32.7 \pm 0.1$ |
| M2A (ours) | $48.7 \pm 0.4$ | $52.8 \pm 0.5$ | $47.7 \pm 0.4$ | $31.8 \pm 1.0$ | $\mathbf{41.3 \pm 1.2}$ | $29.4 \pm 0.9$ |
| M2A (ours, Thai only) | $\mathbf{50.1 \pm 0.2}$ | $\mathbf{53.1 \pm 0.3}$ | $\mathbf{49.4 \pm 0.3}$ | $30.6 \pm 0.8$ | $39.1 \pm 0.3$ | $28.5 \pm 0.9$ |

still improving mathematical correctness. In effect, M2A learns mathematical reasoning without sacrificing multilingual fidelity, whereas other methods either fail to learn reasoning (GRPO) or lose multilingualism (SFT). Appendix E further examines a variant of M2A trained with translation into a single language instead of multiple languages, and detailed per-language results are provided in Appendix F.

## 6.3 FACTUAL REASONING

Table 3 summarizes the performance of the base model (Qwen-2.5-Instruct) and the gains obtained after post-training with GRPO, supervised fine-tuning (SFT), and M2A on GEOFACT-X. For comparison, we also illustrate the performance of other pretrained LLMs (Cohere et al., 2025; Guo et al., 2025). We report reasoning score and answer accuracy. Results are additionally split by whether the language is associated with the country (*assoc.*) or not (*non-assoc.*); for instance, Thai is associated with Thailand but not with the USA. All pretrained models perform better in associative settings, likely because pretraining corpora contain more paired examples where language and country co-occur. This gap underscores the challenge of aligning reasoning across languages and contexts, motivating methods that explicitly enforce language consistency.

M2A achieves the strongest reasoning performance compared to both pretrained and post-trained baselines, and it is reinforced when only using Thai for translation. Notably, M2A improves both settings at a similar rate (4–6% in reasoning score and 20–21% in answer accuracy). A per-language and per-country breakdown is provided in Appendix G. Figure 12 further illustrates model outputs: although all systems reason in the question language (Swahili), only M2A predicts the correct answer.

Finally, we apply machine translation as a post-hoc strategy. Table 4 shows that translation via Google Translate offers no substantial improvements over the direct setting (Table 3), reflecting their weaker multilingual alignment. This suggests that post-hoc translation provides, at most, a superficial fix and fails to address the core challenge of multilingual reasoning.

Table 5: Contribution of individual reward functions to M2A. The evaluation is performed on GSM8K and MGSM. Bold means the best performance. Lang: Language Consistency, CA: Context Alignment, RA: Reasoning-Step Alignment.

| M2A Variants | | | GSM8K | | | MGSM | | |
|---|---|---|---|---|---|---|---|---|
| Lang | CA | RA | Math. | Lang. | Joint | Math. | Lang. | Joint |
| ✓ | | | $86.9 \pm 0.0$ | $\mathbf{100.0 \pm 0.0}$ | $86.9 \pm 0.0$ | $54.2 \pm 0.1$ | $98.3 \pm 0.1$ | $53.8 \pm 0.1$ |
| ✓ | ✓ | | $84.7 \pm 0.1$ | $\mathbf{100.0 \pm 0.0}$ | $84.7 \pm 0.1$ | $57.8 \pm 0.1$ | $\mathbf{99.5 \pm 0.1}$ | $57.5 \pm 0.1$ |
| ✓ | ✓ | ✓ | $\mathbf{87.3 \pm 0.1}$ | $\mathbf{100.0 \pm 0.0}$ | $\mathbf{87.3 \pm 0.1}$ | $\mathbf{59.0 \pm 0.3}$ | $97.8 \pm 0.2$ | $\mathbf{58.1 \pm 0.4}$ |

Table 6: Comparison of reward formulation for multilingual alignment rewards of M2A. The evaluation is performed on GSM8K and MGSM. Bold means the best performance.

| Reward Formulation | GSM8K | | | MGSM | | |
|---|---|---|---|---|---|---|
| | Math. | Lang. | Joint | Math. | Lang. | Joint |
| $\cos(z_o, z_y)$ | $84.1 \pm 0.1$ | $100.0 \pm 0.0$ | $84.1 \pm 0.1$ | $57.4 \pm 0.6$ | $97.4 \pm 0.1$ | $56.0 \pm 0.2$ |
| $\cos(z_o, z_y) - \cos(\tilde{z}_o, \tilde{z}_y)$ | $83.6 \pm 0.1$ | $100.0 \pm 0.0$ | $83.6 \pm 0.1$ | $57.6 \pm 0.1$ | $\mathbf{99.6 \pm 0.1}$ | $57.4 \pm 0.1$ |
| $\max(\cos(z_o, z_y) - \cos(\tilde{z}_o, \tilde{z}_y) + \alpha, 0)$ (ours) | $\mathbf{87.3 \pm 0.1}$ | $\mathbf{100.0 \pm 0.0}$ | $\mathbf{87.3 \pm 0.1}$ | $\mathbf{59.0 \pm 0.3}$ | $97.8 \pm 0.2$ | $\mathbf{58.1 \pm 0.4}$ |

## 6.4 ABLATION STUDY

**Contribution of Individual Reward Functions.** We analyze the effectiveness of individual reward functions in M2A on the mathematical reasoning task. Table 5 shows that context alignment (CA) improves multilingual performance on MGSM but slightly lowers GSM8K accuracy, as enforcing global embedding similarity adds constraints unnecessary for English-only tasks. Reasoning-step alignment (RA) provides finer supervision by aligning individual reasoning steps, which boosts multilingual performance and mitigates the small degradation from CA. The full model, combining language consistency, CA, and RA, achieves the best results on both benchmarks, confirming that the reward functions are complementary: CA promotes global cross-lingual alignment, while RA enforces stepwise reasoning fidelity.

**Reward Formulations.** We compare different formulations of the multilingual alignment reward used in Eq. (2) and Eq. (4). Table 6 reports results on GSM8K and MGSM. Using vanilla cosine similarity yields weaker performance, while adding a negative-sample term improves MGSM but slightly reduces GSM8K. Our margin-based hinge formulation achieves the best results across all metrics, demonstrating the benefit of combining negative samples with a margin to stabilize alignment.

## 7 DISCUSSION

We conducted a comprehensive study of whether large language models (LLMs) reason in the language of the input question. Our findings show that many LLMs predominantly reason in English or Chinese, even when prompted in other languages, undermining multilingual reasoning quality and limiting their applicability in culturally and linguistically diverse settings.

To overcome this limitation, we introduce a novel method, M2A, which enforces language-consistent reasoning while preserving factual correctness. By combining multi-scale alignment rewards with a language-consistency objective, M2A aligns outputs with ground-truth reasoning traces at both context and reasoning step levels, encouraging reasoning to remain in the query language.

Robust evaluation of multilingual reasoning is itself difficult, since most benchmarks focus on final answers rather than reasoning quality or language alignment. We therefore propose GEOFACT-X, a geography-based factual reasoning benchmark spanning five diverse languages, paired with step-by-step reasoning traces and a reasoning evaluation protocol including logical structure, factual correctness, and language consistency.

Our results show that M2A consistently improves multilingual mathematical and factual reasoning capability while maintaining strong English performance. While our experiments were conducted on 7B-parameter models, the approach is scalable and provides a practical alternative to massive multilingual instruction tuning. More broadly, our contributions establish a foundation for training and evaluating LLMs that reason faithfully across languages, advancing the goal of globally inclusive, culturally grounded, and interpretable AI.

## ETHICS STATEMENT

We read the ICLR Code of Ethics before the submission. Our paper focuses on multilingual reasoning capabilities of large language models (LLMs), emphasizing knowledge transfer between languages and highlighting limitations faced by low-resource languages. We believe this work encourages the research community to address these limitations, ultimately contributing toward equitable access to high-performing LLMs, regardless of the user's language. However, our work also shares similar negative societal concerns with standard large language model research (*e.g.*, biased toward high-resource languages and hallucination).

## REPRODUCIBILITY STATEMENT

We specify the experimental setting in Appendix B and attached the codebase as supplementary materials.

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

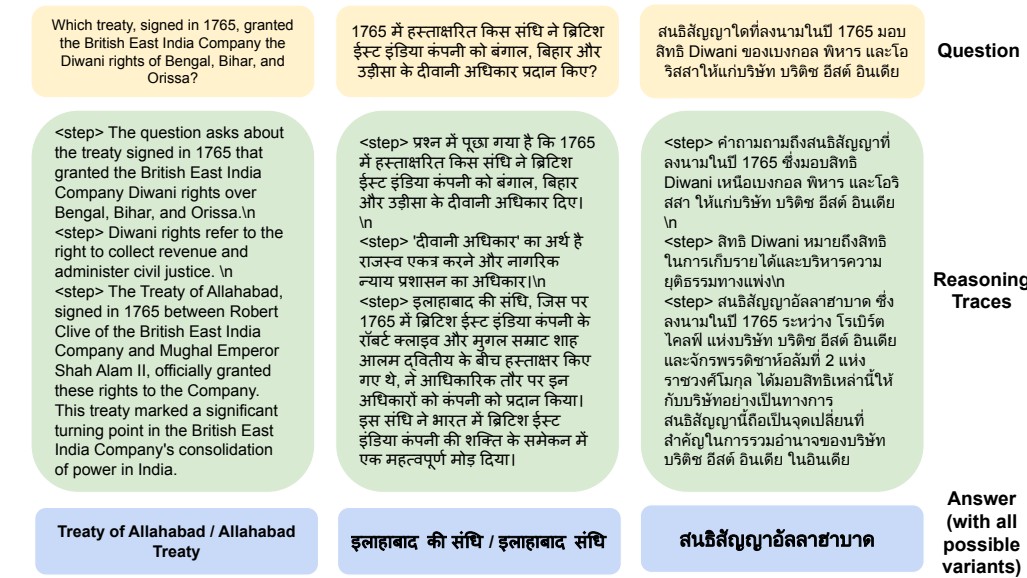

Figure 6: **A sample from GEOFACT-X in English, Hindi, and Thai.** Each presents the same factual question and answer content translated across languages. These multilingual and semantically equivalent traces serve as reference reasoning for benchmarking the reasoning quality of other language models in our evaluation framework.

## A  DETAILS OF GEOFACT-X

### A.1  DATASET COLLECTION

We constructed a multilingual factual QA dataset using Gemini 2.0 Flash. For each country–language pair (USA–English, India–Hindi, Japan–Japanese, Kenya–Swahili, Thailand–Thai), we generated 600 unique QA pairs (3,000 examples in total) by using prompt templates shown in Figure 7. The topics spanned ten high-level domains: History, Geography, Politics, Literature, Arts & Culture, Science & Technology, Sports, Food & Cuisine, Language, and Religion with subcategories such as *Person*, *Date*, and *Place* (Figure 8). For each subcategory, 20 questions were generated per country. Translations were produced with Google Translate, and semantic fidelity was checked via back-translation. Reasoning traces were generated by Gemini 2.0 Flash using Chain-of-Thought prompting (Fig. 9), with each step explicitly tagged by a '<step>' token.

The dataset is split into training (85%) and test (15%) sets, with no semantic overlap across splits or languages. Ten percent of the training data and all test data were manually verified by the authors through cross-referencing with Wikipedia and Google Search. In addition, all test samples were reviewed by native or C1-level speakers to ensure factual correctness and linguistic clarity and modify the samples if needed.

### A.2  EVALUATION PROTOCOL

The benchmark has three metrics, answer accuracy, and reasoning score. Answer accuracy is computed by checking whether the model prediction appears in the list of reference answers provided for each test instance. Reasoning score is evaluated with Qwen-2.5-72B-Instruct (Qwen et al., 2025) as an LLM-as-a-Judge, which compares model-generated reasoning traces against human-verified references to measure the quality of generated reasoning. If a reasoning trace is written in a different language from the question, detected by a language identifier (Lui & Baldwin, 2012), its score is set to zero. Figure 10 illustrates the prompt structure used for the LLM-as-a-Judge, including the evaluation instructions and rules applied to model outputs.

Generate {num_questions} factual questions about {country} focused on the topic of {topic} where the answer type is {answer_type}.
Requirements:
    1. Each question must have a SINGLE, DEFINITE answer (not subjective or opinion-based).
    2. Focus on facts that are well-established and locally known in {country}.
    3. For each answer, provide ALL possible correct variants (e.g., full names, common abbreviations, alternative names).
    4. DO NOT include any ambiguous questions where the answer could be interpreted in multiple ways.
    5. Each question should be translated into exactly these languages: English, Hindi, Japanese, Swahili, and Thai.

    CRITICAL TRANSLATION REQUIREMENTS:
    - Ensure HIGHEST QUALITY translations in all languages. Translations must be accurate and natural-sounding.
    - For proper nouns, provide BOTH the transliterated version AND the commonly accepted translation in each language.
    - Pay special attention to terms that have specific cultural meaning or context.
    - Maintain consistent terminology across all translations of the same question/answer.
    - For Hindi translations: Follow modern standard Hindi conventions and proper transliteration standards.
    - For Japanese translations: Use appropriate kanji, hiragana, and katakana. Include both kanji and phonetic readings where appropriate.
    - For Swahili translations: Use standard Swahili spelling and grammar conventions.
    - For Thai translations: Use proper Thai script and formal Thai language.
    - When translating names of people, places, or specific terms, include commonly recognized translations in each language.
    6. For EACH language version, provide ALL possible correct answer variants in that language.
    7. Questions should be DIVERSE within the selected topic - avoid redundant or very similar questions.
    8. Ensure the answers are SPECIFIC and PRECISE - avoid phrases or long explanations as answers.

    Return the data in the following JSON format:
    {{
      "question_id": "unique_incrementing_number",
      "languages": [
      {{
      "language_code": "en",
      "language_name": "English",
      "question": "The exact question text in English",
      "answers": ["Primary answer", "Alternative form 1", "Alternative form 2"]
      }},
      {{
      "language_code": "hi",
      "language_name": "Hindi",
      "question": "The exact question text in Hindi",
      "answers": ["Primary answer in Hindi", "Alternative form 1 in Hindi", "Alternative form 2 in Hindi"]
      }},
      // Repeat for Japanese (ja), Swahili (sw), and Thai (th)
    ],
    "topic": "{topic}",
    "answer_type": "{answer_type}",
    "region": "{country}"
    }}
    IMPORTANT: Return ONLY valid JSON without any explanations, formatting, or additional text outside the JSON structure. Ensure all apostrophes, quotation marks, and special characters are properly escaped in the JSON.

Figure 7: **Prompt for Generating Multilingual Factual Questions and Answers in GEOFACT-X.** This prompt instructs the LLM to generate diverse, unambiguous factual questions about a specific country and topic, each with a single, definite answer. The questions and their answers are provided in five languages, English, Hindi, Japanese, Swahili, and Thai, with strict requirements for high-quality translations, consistent terminology, and inclusion of all valid answer variants in each language.

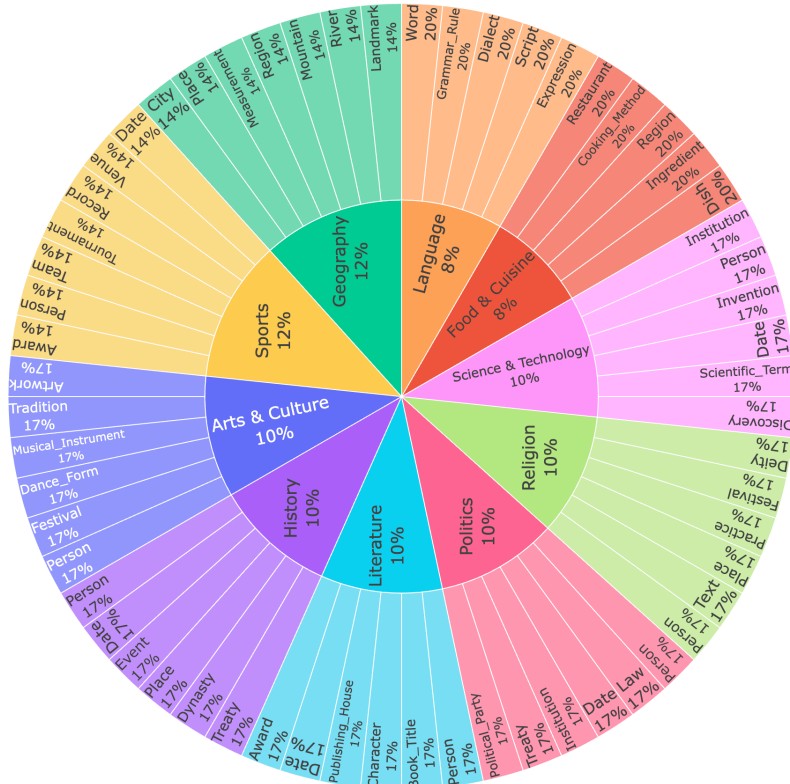

Figure 8: **Illustration of the hierarchical distribution of generated factual question categories by topic and subcategory.** Each colored wedge represents a major topic (e.g., History, Geography), and its outer segments represent specific subcategories (e.g., Person, Place, Treaty). The size of each segment reflects the proportion of questions allocated to that subcategory within its topic. This generation schema was applied uniformly across five countries, and all question sets were translated into five different languages.

## A.3 LICENSE

We release the dataset under the MIT License, which permits reuse, modification, and distribution, provided that the original license and copyright notice are included.

## B EXPERIMENTAL DETAILS

### B.1 TRAINING

We adopt the hyperparameter configuration from s1 (Muennighoff et al., 2025), with the exception of batch size, which we reduced from 16 to 8 due to resource constraints. Specifically, the training hyperparameters are as follows: learning rate of $10^{-5}$, minimum learning rate of 0, weight decay of $10^{-4}$, total batch size of 8, training conducted for 5 epochs, and a cosine learning rate scheduler with warmup ratio of 0.05. Adam (Kingma, 2015) is used with $\beta_1 = 0.9$ and $\beta_2 = 0.95$. The maximum sequence and token lengths are set to 20,000. GRPO uses accuracy and format rewards following Guo et al. (2025). M2A uses a maximum completion length of 1,024 (256) tokens, generates 2 (8) completions per prompt, and sets the maximum step to 10 due to the resource constraints (parentheses denote factual reasoning parameters in case of difference). We also use a loss coefficient for GRPO as 0.01 for mathematical reasoning and 0.5 for factual reasoning. We train the model with three different random seeds to calculate the standard error. For the s1K-X dataset, we use Google Translate to translate the s1K-1.1 dataset into multiple languages used in the MGSM benchmark: Bengali,

```
You are a multilingual reasoning assistant. For each of the following factual questions about the country,
generate structured output in the following format for all the 5 questions:
{
"question": "",
"answer_list": [""],
"reasoning": "<step> Step-by-step reasoning in the same language as the question, each step starting
with <step>",
"corrected_answer_list": [""],
"topic": "",
"region": "",
"answer_type": ""
}
Instructions:
1. The reasoning must be written strictly in the same language as the question.
2. Each reasoning step must begin with <step>
(e.g., <step> माउंट केन्या एक ज् वालामुखी पर्व्वत है ...).
3. Start from relevant background knowledge or interpretation of the question and proceed step-by-step
toward the correct answer.
4. For the corrected_answer_list:
- Review the provided answer_list.
- Remove duplicates (case-insensitive, spacing-normalized).
- Add valid alternative phrasings or translations if any are missing (e.g., transliterations, local variants).
5. The reasoning must be comprehensive and detailed, including:
- Relevant background information and definitions of key terms or entities.
- Historical, cultural, geographical, or scientific context if applicable.
- Logical deductions and connections to prior knowledge.
- Contrast with similar or confusing facts (e.g., common misconceptions).
- Justifications for why incorrect options are incorrect (if multiple answers are possible).
-   Step-by-step elimination or validation of answer candidates.
6. Each <step> should be at least 1–2 full sentences and contribute meaningfully to building up the
answer. Do not skip intermediate steps, even if obvious. Think like a teacher explaining to a curious
student
```

Figure 9: **Structured prompt for multilingual factual reasoning generation using Gemini 2.0 Flash on GEOFACT-X.** This prompt guides the model to generate step-by-step reasoning and corrected answers for factual questions about a country, using the same language as the input question. The output consists of five JSON object strings for the same factual question, each in a different language.

German, Spanish, French, Japanese, Russian, Swahili, Telugu, Thai, and Chinese. Both codebase and datasets are attached to the supplementary material for reproducibility.

## B.2 EVALUATION

For mathematical reasoning, we employed the lm-evaluation-harness[1] library to evaluate each model. Specifically, we used the MGSM (Shi et al., 2023) Native-CoT setting and the MMLU-ProX (Xuan et al., 2025) Math category with a 5-shot chain-of-thought prompt to ensure the model reasons in its native language. langid (Lui & Baldwin, 2012) is used to evaluate language correctness.

## B.3 MODELS EVALUATED ON MGSM

Table 7 provides the complete list of models evaluated in Figure 5. All models are sourced from HuggingFace[2], a public repository of large language models. We include Qwen2.5 (Qwen et al., 2025), s1 (Muennighoff et al., 2025), Llama (Grattafiori et al., 2024), Gemma (Team et al., 2025), and DeepSeek-R1 (Guo et al., 2025), each with a range of model sizes.

---

[1]https://github.com/EleutherAI/lm-evaluation-harness
[2]https://huggingface.co/

```
# Reasoning Quality Evaluation
 You are an expert reasoning evaluator tasked with comparing an LLM's reasoning trace against a ground
truth reasoning trace. Your evaluation must be fair, consistent, and based solely on the quality of
reasoning, not on superficial similarities.

 ## Input:
   - Question: {question}
   - Answer List: {answer_list}
   - Ground Truth Reasoning: {ground_truth_reasoning}
   - LLM Response: {llm_generation}

 ## Evaluation Criteria:
 Assess the quality of the LLM's reasoning compared to the ground truth on a scale from 0-10 based on
the following:

   1. Logical Structure (40%):
      - How well does the reasoning follow a clear, step-by-step logical progression?
       - Are the steps in a sensible order that builds toward the answer?

   2. Key Insights (30%):
      - Does the reasoning identify the same critical insights as the ground truth?
      - Are the important clues from the question properly recognized and utilized?

   3. Factual Correctness (20%):
      - Is the reasoning free from factual errors?
      - Does it avoid adding irrelevant information or missing necessary information?

   4. Conclusion Validity (10%):
      - Does the reasoning correctly lead to the answer?
      - Is the link between the reasoning and the conclusion clear?

 ## Scoring Guide:
   0-1: Completely irrelevant or fundamentally flawed reasoning
   2-3: Major logical errors or missing critical insights
   4-5: Contains some correct elements but misses important aspects
   6-7: Good reasoning with minor gaps or imperfections
   8-9: Very good reasoning, almost matching ground truth quality
   10: Perfect reasoning, capturing all key insights with proper structure

 ## Your Response (FORMAT STRICTLY REQUIRED):
   REASONING_SCORE: [integer between 0-10]
   JUSTIFICATION: [Brief explanation of your evaluation, highlighting strengths and weaknesses]
```

Figure 10: **Prompt for LLM-as-a-Judge to Evaluate Reasoning Traces Using Gemini 2.0 Flash.** This prompt guides the evaluation of an LLM-generated reasoning trace against a ground truth using specific criteria such as logical structure, key insights, factual correctness, and conclusion validity. The evaluation is performed by Qwen2.5-72B Instruct, acting as the LLM-as-judge, and includes scoring, language mismatch detection, and answer validation.

## C  ABLATION STUDY OF TRANSLATION METHOD

To ensure the accessibility and scalability of our pipeline, we prioritize computational efficiency and broad language coverage. We select Google Translate as our primary translation tool because it supports over 100 languages—including many low-resource ones—and remains computationally efficient. This design choice allows researchers to reproduce our method without requiring access to expensive LLM APIs or high-end GPUs.

To validate this choice, we conduct an ablation study comparing Google Translate against LLM-based translation methods (specifically GPT-4.1 and GPT-5). As shown in Table 8, we observe no statistically significant difference in downstream model performance between the translation methods.

Table 7: List of all models and sizes evaluated on MGSM in Figure 5. All models are sourced from HuggingFace.

| Model Name | Model Sizes |
|---|---|
| Qwen2.5 | 1.5B, 3B, 7B, 14B, 32B |
| Qwen2.5-Instruct | 1.5B, 3B, 7B, 14B, 32B |
| Qwen2.5-Instruct-GPTQ-Int4 | 1.5B, 3B, 7B, 14B, 32B |
| Qwen2.5-Instruct-GPTQ-Int8 | 1.5B, 3B, 7B, 14B, 32B |
| Qwen2.5-Instruct-AWQ | 7B, 14B, 32B |
| Qwen2.5-Instruct-1M | 7B, 14B |
| Qwen2.5-Math | 1.5B, 7B, 72B |
| Qwen2.5-Math-Instruct | 1.5B, 7B, 72B |
| s1 | 1.5B, 3B, 7B, 14B, 32B |
| Llama-3-Instruct | 8B, 70B |
| Llama-3.3-Instruct | 70B |
| Gemma-3-PT | 1B, 4B, 12B, 27B |
| Gemma-3-IT | 1B, 4B, 12B, 27B |
| DeepSeek-R1-Distill-Qwen | 1.5B, 3B, 7B, 14B, 32B |
| DeepSeek-R1-Distill-Llama | 8B, 70B |

Table 8: Comparison of model performance with different translation methods on average reasoning score (%), language accuracy (%), and answer accuracy (%) on GEOFACT-X test set, evaluated across all examples and split by whether the language is associated with the country ('Assoc.') or not ('Non-Assoc.'). Bold means the best performance.

| Translation Method | Average Reasoning Score (%) | | | Average Answer Accuracy (%) | | |
|---|---|---|---|---|---|---|
| | All | Assoc. | Non-Assoc. | All | Assoc. | Non-Assoc. |
| Google Translate | $48.5 \pm 0.4$ | $\mathbf{52.6 \pm 0.6}$ | $47.5 \pm 0.3$ | $32.0 \pm 0.6$ | $41.3 \pm 1.0$ | $29.7 \pm 0.5$ |
| ChatGPT 4.1 | $48.8 \pm 0.1$ | $52.2 \pm 0.3$ | $48.0 \pm 0.2$ | $\mathbf{33.1 \pm 0.3}$ | $41.3 \pm 1.3$ | $\mathbf{31.0 \pm 0.4}$ |
| ChatGPT 5 | $\mathbf{49.0 \pm 0.1}$ | $\mathbf{52.6 \pm 0.5}$ | $\mathbf{48.1 \pm 0.0}$ | $32.7 \pm 0.3$ | $\mathbf{41.4 \pm 1.0}$ | $30.5 \pm 0.2$ |

These results confirm that our framework is robust to the choice of translator and that the performance gains stem principally from the M2A objective rather than translation artifacts.

## D M2A WITH DIFFERENT BACKBONE ON GEOFACT-X

We use Command R7B (Cohere et al., 2025) as a backbone network for comparing M2A other baselines. As shown in Table 9, the results are consistent with our main Qwen findings. M2A achieves the highest performance in both reasoning score and answer accuracy, outperforming SFT and standard GRPO. This confirms that our method is model-agnostic and effective across different multilingual architectures.

## E M2A WITH DIFFERENT LANGUAGES

In the main paper, we present M2A trained with random translations drawn from the ten MGSM languages (Bengali, German, Spanish, French, Japanese, Russian, Swahili, Telugu, Thai, and Chinese). Here, we examine the effect of using a single fixed translation language, as shown in Table 10. We select Japanese and Swahili as representative examples of high- and low-resource languages, respectively, following the categorization of Nicholas & Bhatia (2023).

Across both choices, we observe only minor decreases in GSM8K mathematical accuracy and MGSM language accuracy relative to the multi-language setting. This finding indicates that training with a single language—even a low-resource one—can still induce strong multilingual reasoning ability. Nevertheless, randomizing translations across multiple languages yields the strongest overall

Table 9: Comparison of model performance on average reasoning score (%), language accuracy (%), and answer accuracy (%) on GEOFACT-X test set, evaluated across all examples and split by whether the language is associated with the country ('Assoc.') or not ('Non-Assoc.'). Bold means the best performance.

| Model | Average Reasoning Score (%) | | | Average Answer Accuracy (%) | | |
|---|---|---|---|---|---|---|
| | All | Assoc. | Non-Assoc. | All | Assoc. | Non-Assoc. |
| Command R7B | 44.2 | 48.5 | 43.1 | 25.0 | 34.1 | 22.7 |
| GRPO | $38.3 \pm 0.4$ | $42.4 \pm 0.9$ | $37.2 \pm 0.4$ | $23.4 \pm 3.2$ | $30.5 \pm 4.2$ | $21.6 \pm 3.0$ |
| SFT | $46.6 \pm 0.0$ | $\mathbf{51.3 \pm 0.3}$ | $45.4 \pm 0.1$ | $28.9 \pm 0.3$ | $38.2 \pm 0.6$ | $26.6 \pm 0.2$ |
| SFT + GRPO | $44.1 \pm 1.4$ | $48.8 \pm 1.9$ | $42.9 \pm 1.2$ | $21.9 \pm 6.8$ | $29.0 \pm 8.9$ | $20.1 \pm 6.3$ |
| M2A | $\mathbf{46.8 \pm 0.8}$ | $50.1 \pm 0.9$ | $\mathbf{46.0 \pm 0.7}$ | $\mathbf{30.9 \pm 0.9}$ | $\mathbf{38.5 \pm 1.3}$ | $\mathbf{29.1 \pm 0.8}$ |

Table 10: Accuracy of M2A trained with various languages. All languages denote that the language translator randomly translates the question language into ten different languages used in MGSM. Bold denotes the best performance for each metric.

| Translation Language | GSM8K | | | MGSM | | |
|---|---|---|---|---|---|---|
| | Math. | Lang. | Joint | Math. | Lang. | Joint |
| All | $\mathbf{87.3 \pm 0.1}$ | $\mathbf{100.0 \pm 0.0}$ | $\mathbf{87.3 \pm 0.1}$ | $\mathbf{59.0 \pm 0.3}$ | $\mathbf{97.8 \pm 0.2}$ | $\mathbf{58.1 \pm 0.4}$ |
| Japanese | $86.8 \pm 0.1$ | $100.0 \pm 0.0$ | $86.8 \pm 0.1$ | $59.0 \pm 0.2$ | $90.7 \pm 0.4$ | $56.5 \pm 0.1$ |
| Swahili | $85.3 \pm 0.7$ | $100.0 \pm 0.0$ | $85.3 \pm 0.7$ | $58.4 \pm 0.3$ | $96.2 \pm 1.0$ | $56.6 \pm 0.7$ |

Table 11: Accuracy of base (Qwen2.5-7B-Instruct) model and models fine-tuned with each post-training method on MGSM. Standard error is not included for readability. Bold means the best performance.

| Model | MGSM | Question Language | | | | | | | | | |
|---|---|---|---|---|---|---|---|---|---|---|---|
| | | bn | de | es | fr | ja | ru | sw | te | th | zh |
| Math Performance | | | | | | | | | | | |
| Qwen2.5-Instruct | 58.7 | 61.2 | 72.0 | 72.8 | 62.4 | 70.4 | 65.6 | 14.0 | 29.6 | 69.6 | 69.2 |
| GRPO | 58.8 | 59.7 | 69.7 | 75.9 | 65.7 | 66.9 | 71.7 | 12.8 | 27.7 | 64.9 | 72.4 |
| SFT (s1) | $\mathbf{66.7}$ | $\mathbf{66.4}$ | $\mathbf{77.4}$ | $\mathbf{78.0}$ | $\mathbf{79.6}$ | $\mathbf{73.6}$ | $\mathbf{83.6}$ | $\mathbf{18.6}$ | $\mathbf{35.2}$ | $\mathbf{74.6}$ | $\mathbf{79.8}$ |
| SFT on s1K-X | 45.2 | 34.7 | 51.2 | 60.8 | 69.3 | 44.0 | 52.5 | 10.4 | 8.8 | 64.7 | 55.1 |
| M2A | 59.0 | 53.3 | 75.6 | 75.2 | 75.7 | 66.3 | 80.5 | 3.7 | 14.9 | 66.5 | 78.3 |
| Language Performance | | | | | | | | | | | |
| Qwen2.5-Instruct | 99.0 | $\mathbf{100.0}$ | 100.0 | 100.0 | 100.0 | 91.6 | 100.0 | 100.0 | 100.0 | 98.8 | 100.0 |
| GRPO | 95.9 | 99.9 | $\mathbf{100.0}$ | $\mathbf{100.0}$ | 99.9 | $\mathbf{100.0}$ | 100.0 | 60.0 | 100.0 | 99.6 | 100.0 |
| SFT (s1) | 31.0 | 13.6 | 81.4 | 88.6 | 5.8 | 6.0 | 2.6 | 0.0 | 22.8 | 27.8 | 62.2 |
| SFT on s1K-X | $\mathbf{99.7}$ | 100.0 | 99.9 | 99.9 | 99.9 | $\mathbf{100.0}$ | 99.7 | 99.9 | $\mathbf{100.0}$ | $\mathbf{100.0}$ | 98.0 |
| M2A | 97.8 | 99.7 | 98.0 | 98.5 | 94.8 | 99.9 | 99.2 | 88.3 | 99.9 | 99.3 | $\mathbf{100.0}$ |
| Joint Performance | | | | | | | | | | | |
| Qwen2.5-Instruct | 58.1 | $\mathbf{61.2}$ | 72.0 | 72.8 | 62.4 | 65.2 | 65.6 | $\mathbf{14.0}$ | $\mathbf{29.6}$ | $\mathbf{68.8}$ | 69.2 |
| GRPO | $\mathbf{58.2}$ | 59.6 | 69.7 | 75.9 | 65.6 | 66.9 | 71.7 | 7.7 | 27.7 | 64.5 | 72.4 |
| SFT (s1) | 21.9 | 5.6 | 62.8 | 68.6 | 4.6 | 2.4 | 1.2 | 0.0 | 5.8 | 18.6 | 49.0 |
| SFT on s1K-X | 45.0 | 34.7 | 51.2 | 60.8 | 69.3 | 44.0 | 52.5 | 10.4 | 8.8 | 64.7 | 53.7 |
| M2A | 58.1 | 53.2 | $\mathbf{74.1}$ | $\mathbf{74.3}$ | $\mathbf{71.2}$ | 66.3 | $\mathbf{80.4}$ | 2.7 | 14.9 | 66.0 | $\mathbf{78.3}$ |

results, suggesting that language diversity provides additional regularization benefits for cross-lingual alignment.

## F  MGSM EVALUATION IN EACH LANGUAGE

Table 11 shows individual math and language accuracy change compared to the base model (Qwen2.5-7B-Instruct) in each language. As we mentioned in Section 6, supervised fine-tuning on s1K-1.1

Table 12: Average reasoning score (%) by language and region. Reasoning quality is assessed using an LLM-as-a-judge framework, which evaluates model-generated justifications against reference Gemini 2.0 Flash reasoning traces in the GEOFACT-X dataset. Higher scores indicate more coherent, relevant, and logically sound reasoning. The gray diagonal entries represent associated language–country pairs. Bold means the best performance in each pair.

| Language | Model | USA | India | Japan | Kenya | Thailand |
|---|---|---|---|---|---|---|
| English | Qwen2.5-Instruct | 67.5 | 55.7 | **62.8** | 56.6 | 51.4 |
| | GRPO | **70.3 ± 1.2** | 61.8 ± 1.3 | 58.5 ± 1.2 | 57.2 ± 0.7 | **51.9 ± 0.6** |
| | SFT | 69.2 ± 0.6 | 62.9 ± 0.7 | 57.6 ± 0.6 | 57.4 ± 0.1 | 51.2 ± 0.7 |
| | M2A | 69.3 ± 0.5 | **63.0 ± 0.7** | 57.8 ± 0.7 | **57.4 ± 0.2** | 51.5 ± 0.8 |
| Hindi | Qwen2.5-Instruct | **39.4** | **39.0** | **39.0** | 35.0 | 38.6 |
| | GRPO | 38.7 ± 1.4 | 36.6 ± 0.9 | 38.2 ± 0.7 | 35.5 ± 0.7 | 38.2 ± 0.9 |
| | SFT | 36.3 ± 1.0 | 36.0 ± 0.5 | 37.7 ± 1.4 | 42.4 ± 0.5 | **40.8 ± 0.9** |
| | M2A | 36.5 ± 1.1 | 36.0 ± 0.4 | 38.2 ± 1.1 | **42.5 ± 0.5** | 40.8 ± 0.9 |
| Japanese | Qwen2.5-Instruct | **51.4** | 43.1 | **53.2** | 45.2 | 40.3 |
| | GRPO | 50.1 ± 1.4 | **43.2 ± 0.6** | 52.3 ± 0.1 | 47.0 ± 1.2 | 43.5 ± 0.6 |
| | SFT | 45.3 ± 1.1 | 43.0 ± 0.6 | 52.3 ± 1.3 | 47.7 ± 0.7 | 43.0 ± 0.2 |
| | M2A | 45.5 ± 1.1 | 43.0 ± 0.5 | 52.6 ± 1.4 | **48.1 ± 0.7** | **43.5 ± 0.3** |
| Swahili | Qwen2.5-Instruct | 41.4 | 43.3 | 39.5 | 41.3 | 34.9 |
| | GRPO | 34.6 ± 0.5 | 33.2 ± 1.4 | 31.4 ± 0.6 | 35.6 ± 1.2 | 33.8 ± 0.4 |
| | SFT | 47.9 ± 0.2 | **49.8 ± 1.3** | 47.4 ± 0.2 | 49.6 ± 0.4 | 47.3 ± 0.7 |
| | M2A | **48.0 ± 0.3** | **49.8 ± 1.3** | **47.5 ± 0.2** | **49.7 ± 0.4** | **47.3 ± 0.8** |
| Thai | Qwen2.5-Instruct | **56.4** | 44.0 | **49.0** | 43.5 | 45.3 |
| | GRPO | 54.1 ± 1.4 | **47.5 ± 0.3** | 46.7 ± 1.6 | **47.8 ± 0.6** | 45.0 ± 0.6 |
| | SFT | 40.3 ± 0.8 | 40.2 ± 0.3 | 47.2 ± 0.6 | 45.1 ± 0.1 | 45.5 ± 0.4 |
| | M2A | 40.3 ± 0.6 | 40.5 ± 0.3 | 47.5 ± 0.6 | 45.5 ± 0.2 | **45.6 ± 0.4** |

Table 13: Average answer accuracy by language and region. The gray diagonal entries represent associated language–country pairs. Bold means the best performance in each pair.

| Lang | Model | USA | India | Japan | Kenya | Thailand |
|---|---|---|---|---|---|---|
| English | Qwen2.5-Instruct | 56.3 | 32.7 | 41.9 | 31.6 | 24.4 |
| | GRPO | **74.7 ± 1.1** | **62.9 ± 2.1** | **52.0 ± 2.4** | **49.1 ± 2.1** | **38.4 ± 0.7** |
| | SFT | 70.9 ± 1.9 | 60.2 ± 2.0 | 47.0 ± 1.3 | 45.3 ± 0.6 | 34.5 ± 1.0 |
| | M2A | 65.6 ± 2.2 | 59.0 ± 3.0 | 51.7 ± 2.2 | 43.1 ± 5.6 | 31.1 ± 2.3 |
| Hindi | Qwen2.5-Instruct | 19.5 | **23.3** | **16.1** | 14.3 | 20.0 |
| | GRPO | **22.8 ± 1.5** | 22.1 ± 1.8 | 11.9 ± 2.0 | 13.9 ± 1.0 | **20.8 ± 0.8** |
| | SFT | 11.4 ± 1.6 | 14.7 ± 2.1 | 10.0 ± 0.4 | 18.3 ± 0.4 | 15.0 ± 1.4 |
| | M2A | 15.1 ± 1.8 | 19.7 ± 1.7 | 13.3 ± 1.9 | **19.8 ± 1.2** | 14.5 ± 2.4 |
| Japanese | Qwen2.5-Instruct | **36.8** | 23.0 | 39.3 | 20.2 | 18.9 |
| | GRPO | 34.6 ± 2.7 | **25.3 ± 1.8** | 40.5 ± 2.1 | 29.0 ± 1.4 | **26.6 ± 2.4** |
| | SFT | 32.9 ± 1.3 | 20.3 ± 0.8 | 42.1 ± 4.0 | **29.4 ± 0.8** | 19.8 ± 1.2 |
| | M2A | 30.8 ± 2.6 | 25.1 ± 0.6 | **42.5 ± 2.0** | 27.9 ± 3.1 | 20.3 ± 2.0 |
| Swahili | Qwen2.5-Instruct | 24.3 | 32.6 | 17.6 | 22.6 | 13.5 |
| | GRPO | 29.3 ± 0.9 | 25.6 ± 1.8 | 15.5 ± 0.4 | 19.6 ± 1.7 | 18.9 ± 0.0 |
| | SFT | 32.4 ± 2.1 | **34.5 ± 2.5** | **21.7 ± 0.8** | 26.7 ± 1.4 | **25.7 ± 2.1** |
| | M2A | **33.3 ± 3.5** | 33.3 ± 1.1 | 21.1 ± 1.6 | **28.3 ± 3.4** | 22.2 ± 0.7 |
| Thai | Qwen2.5-Instruct | 31.9 | 23.8 | 18.5 | 20.7 | 25.7 |
| | GRPO | **39.6 ± 1.7** | **29.2 ± 0.8** | **29.2 ± 1.1** | **30.1 ± 2.3** | 29.3 ± 3.2 |
| | SFT | 15.0 ± 0.5 | 19.6 ± 1.1 | 20.6 ± 0.4 | 19.5 ± 0.7 | **29.3 ± 0.5** |
| | M2A | 28.6 ± 3.4 | 23.3 ± 0.0 | 27.8 ± 3.6 | 28.6 ± 2.4 | 24.3 ± 1.8 |

improves math performance while losing language performance. Conversely, GRPO rarely changes performance on both mathematics and language across all languages. M2A generally maintains language performance. However, its German performance is much lower, which might be related to the language sampled for translating the question.

We also illustrate an MGSM Russian example response from Qwen2.5-7B-Instruct and SFT, GRPO, M2A fine-tuned models on Figure 11. All models generate the correct answer from a given question written in Russian. However, the SFT model uses English instead of Russian, while others reason in Russian. GRPO has an almost identical reasoning process to the base model, which might explain why it performs almost the same in any metrics as the base model.

## G  GEOFACT-X EVALUATION IN EACH LANGUAGE AND COUNTRY PAIR

Tables 12, and 13 present reasoning score, language accuracy, and answer accuracy for each language–country pair. Diagonal entries correspond to associative pairs, while off-diagonal entries correspond to non-associative pairs. Associative countries consistently achieve higher performance across all metrics compared to non-associative ones. English (a high-resource language) generally outperforms other languages. Among non-associative cases, performance for the United States exceeds that of Kenya and Thailand, which may reflect differences in pre-training and fine-tuning data exposure. As discussed in Section 6.3, all methods substantially increase language accuracy as the training dataset includes all five languages. However, no consistent trend is observed in reasoning score and answer accuracy across language–country pairs.

## H  HUMAN AGREEMENT FOR LLM-AS-A-JUDGE

We validate the alignment between reasoning score and human judgment through a Human-LLM agreement study on the English-USA subset of GeoFact-X. A human evaluator (native English speaker) performs pairwise A/B testing, selecting the superior reasoning trace from two model outputs and assigning a confidence score on a 1–5 scale. The results demonstrate strong alignment between the Qwen-based judge and human preference, achieving an agreement rate of 82.9%.

We further observe a strong positive correlation between human reported confidence and the reasoning score difference assigned by the LLM. Notably, human confidence is significantly higher when agreeing with the LLM (average confidence 3.3/5) compared to cases of disagreement (average confidence 2.1/5). This indicates that divergences primarily occur in ambiguous "close calls," whereas the metric remains highly reliable for clear-cut distinctions. Future work will expand this validation across all five languages with native speakers to confirm these findings multilingually.

## I  LLM USAGE

We used LLM to help with paper writing for improving grammar and wording.

**Question**

'<|im_start|>system
You are Qwen, created by Alibaba Cloud. You are a helpful assistant.<|im_end|>
<|im_start|>user\nЗадача: Утки Дженет несут по 16 яиц в день. Она ест три на завтрак каждое утро и печет кексы для своих друзей каждый день, используя четыре. Остаток она аждый день продает на фермерском рынке по 2 $ за свежее утиное яйцо. Сколько в долларах она зарабатывает на фермерском рынке каждый день?\nПошаговоерешение:<|im_end|>
<|im_start|>assistant

**Qwen2.5-7B-Instruct**

Давайте решим эту задачу пошагово:
1. Сначала определим, сколько яиц остается после того, как утки Дженет съедают и используют их:
  - Всего яиц в день: 16
  - Яиц на завтрак: 3
  - Яиц для кексов: 4
  - Остаток: 16 - 3 - 4 = 9 яиц

2. Теперь определим, сколько Дженет продает на фермерском рынке:
  - Она продает все оставшиеся яйца: 9 яиц

3. Посчитаем, сколько денег она зарабатывает:
  - Цена за яйцо: 2 $
  - За 9 яиц: 9 * 2 = 18 $

Таким образом, Дженет зарабатывает **18** долларов на фермерском рынке каждый день.

**Supervised Fine-Tuning (s1)**

Okay, let's try to figure out how much money Janet's ducks make each day from selling eggs at the farmer's market. Let me break down the problem step by step.

First, the problem states that Janet's ducks lay 16 eggs each day. That's the total production. Then, Janet eats 3 eggs every morning for breakfast. So, those 3 eggs are consumed by her. Additionally, she uses 4 eggs each day to bake cakes for her friends. So, both her consumption and baking take a total of 3 + 4 = 7 eggs per day.

...

But just to make sure I didn't misread the question. Let me re-read it:

"Утки ••• день?"

Which translates to: "Janet's ducks lay 16 eggs per day. She eats three for breakfast every morning and bakes cakes for her friends using four eggs every day. The remainder she sells at the farmer's market for $2 per fresh duck egg each day. How much does she earn at the farmer's market each day in dollars?"

Yes, the question is asking about her daily earnings from the farmer's market. So, the calculation is correct.

Therefore, the final answer is \boxed{18}.
<|im_start|>answer
Answer: Janet's ducks lay 16 eggs each day. She eats 3 eggs every morning and uses 4 eggs to bake cakes for her friends. The total number of eggs used each day is 3 + 4 = 7. The remaining eggs are 16 - 7 = 9. She sells each egg for $2, so her daily earnings are 9 * $2 = $18.

**\boxed{18}**

**GRPO**

Давайте решим эту задачу пошагово:

1. Сначала определим, сколько яиц остается после того, как утки Дженет съедают и используют их для кексов:
  - Всего яиц в день: 16
  - Яиц, съеденных на завтрак: 3
  - Яиц, использованных для кексов: 4
  - Остаток яиц: 16 - 3 - 4 = 9

2. Теперь определим, сколько денег Дженет зарабатывает на продаже оставшихся яиц:
  - Цена за одно яйцо: 2 $
  - Количество продаваемых яиц: 9
  - Общая сумма за день: 9 * 2 = 18 $

Таким образом, Дженет зарабатывает **18** долларов на фермерском рынке каждый день.

**M2A**

Okay, давайте решим эту задачу шаг за шагом. У нас есть информация о том, что утки Дженет несут 16 яиц в день. Она сама ест три яйца на завтрак каждый день и использует еще четыре яйца для печения кексов для своих друзей. Остаток яиц она продает на фермерском рынке по 2 доллара за каждое свежее утиное яйцо. Нам нужно определить, сколько она зарабатывает на фермерском рынке каждый день.

**Шаг 1: Вычислить количество яиц, которые Дженет использует на свой завтрак и для кексов.**

Дженет ест 3 яйца на завтрак каждый день. Кроме того, она использует 4 яйца для кексов для друзей каждый день. Всего яиц, которые она использует сама или дает друзьям, будет 3 + 4 = 7 яиц в день.

**Шаг 2: Вычислить количество оставшихся яиц, которые она может продать.**

Утки несут 16 яиц в день. Если из них 7 яйца используются Дженет и ее друзьями, то оставшихся будет 16 - 7 = 9 яиц. Эти 9 яиц она продает на фермерском рынке.

**Шаг 3: Вычислить доход от продажи оставшихся яиц.**

Каждое яйцо продаётся по 2 доллара. Поэтому, если она продает 9 яиц по 2 доллара каждое, то доход составит 9 * 2 = 18 долларов.

**Ответ:** Дженет зарабатывает 18 долларов на фермерском рынке каждый день.

**Проверка:** Давайте проверим свои вычисления. Яйца в день: 16. Используется на завтрак и кексы: 3 + 4 = 7. Осталось: 16 - 7 = 9. Цена: 9 * 2 = 18. Все верно. Ответ: 18 долларов.

**Конечный ответ:** \boxed{18}
<|im_start|>answer
Ответ: Утки Дженет несут 16 яиц в день. Она использует 3 яйца на завтрак и 4 яйца для кексов, что в сумме составляет 3 + 4 = 7 яиц. Остаток яиц, который она продает на фермерском рынке, составляет 16 - 7 = 9 яиц. По 2 долларов за каждое яйцо она зарабатывает 9 * 2 = 18 долларов в день.

**\boxed{18}**

Figure 11: **Example question and response of each model on MGSM Russian.** The question in English is "Janet's ducks lay sixteen eggs per day. She eats three for breakfast every morning and bakes muffins for her friends every day with four. She sells the remainder at the farmers' market daily for two dollars per fresh duck egg. How much in dollars does she make every day at the farmers' market?" All models correctly provide the answer (18), but the SFT model conducts its reasoning in English. GRPO's reasoning process is almost identical to that of the base model (Qwen2.5-7B-Instruct).

| Model | Q: Safari ya Lewis na Clark ilianza wapi? (Where did Lewis and Clark's journey begin?) A: St. Louis |
|---|---|
| Base | Swali linauliza kuhusu mahali ambapo safari ya Lewis na Clark ilianza.
>Safari ya Lewis na Clark ilikuwa safari ya kuvuka juu ya maji ya Kimarekani. Ililianza mnamo 1804 na ilishuhudia kipindi cha kwanza cha Marekani kushoto mashariki.
>Safari hii ilianza Fort Atkinson, ambayo ilikuwa fortuoni ya watalii huko Council Bluffs, Iowa, na Omaha, Nebraska. Kwa hivyo, majibu yote mawili yanakubalika.
<answer>Fort Atkinson</answer>

**Translated Output:**
The question asks where the Lewis and Clark expedition began.
>The Lewis and Clark expedition was a voyage across American waters. It began in 1804 and marked the first American expedition eastward.
>The expedition began at Fort Atkinson, which was a tourist fort in Council Bluffs, Iowa, and Omaha, Nebraska. Therefore, both answers are acceptable.
<answer>Fort Atkinson</answer> |
| GRPO | Safari ya Lewis na Clark ilianza wapi?
Hii inatafsiriwa kama safari ya kawaida kwa ajili ya kujenga kazi kwa kawaida na kujenga mazingira na kawaida.
Iliopiga safari hili, Lewis na Clark inatafsiriwa kama kawaida kwa ajili ya kujenga mazingira na kawaida na kujenga kazi kwa kawaida.
Safari hili ina kujenga mazingira na kawaida kwa ajili ya kujenga kazi kwa kawaida.
<answer> Safari ya Lewis na Clark ilianza kwa ajili ya kujenga mazingira na kawaida na kujenga kazi kwa kawaida. </answer>

**Translated Output:**
The Lewis and Clark expedition began in 1804 and 1806.
To help with births and births.
To learn if there are countries with births and births and births and long periods, Lewis and Clark began in 1804 and 1806.
To learn if there are countries with births and births and long periods, Lewis and Clark began in 1804 and 1806.
<answer> The Lewis and Clark expedition began in 1804 and 1806. </answer> |
| SFT | Swali linauliza kuhusu mahali ambapo safari ya Lewis na Clark ilianza. Safari hii ilikuwa safari ya kuvuka juu ya maji ya Katika Marekani baada ya ununuzi wa Louisiana.
<step> Meriwether Lewis alikuwa mwanamke mtawalia ambaye aliongoza safari ya Lewis na Clark. Alikuwa pia vice presidenta wa Theodore Roosevelt.
<step> Safari ya Lewis na Clark ilianza Washington, D.C. Washington, D.C. ndio eneo kuu la bandari nchini Marekani na linajulikana kwa kuwa kituo cha kikatiba cha Marekani.
<step> Jiji la Washington ni jina lingine la Washington, D.C., hivyo majibu yote mawili ni sahihi.
<answer>Washington, D.C.</answer>

**Translated Output:**
The question asks where the Lewis and Clark expedition began. This expedition was a journey across the waters of the United States after the Louisiana Purchase.
<step> Meriwether Lewis was a woman who led the Lewis and Clark expedition. She was also Theodore Roosevelt's vice president.
<step> The Lewis and Clark expedition began in Washington, D.C. Washington, D.C. is the largest port in the United States and is known for being the constitutional capital of the United States.
<step> The city of Washington is another name for Washington, D.C., so both answers are correct.
<answer>Washington, D.C.</answer> |
| M2A | Swali linauliza kuhusu mahapo safari ya Lewis na Clark ilianza.
Safari ya Lewis na Clark ilikuwa safari ya kuvuka juu ya ardhi ya Amerika Kaskazini ili kuashiria maji ya Missouri na kubadilisha mapema ya Marekani.
Ilianzishwa mwaka 1804 na ilishirikisha Merikani.
Utafiti wa kihistoria na kuvunjika kwenye muktadha wa safari huo unaonyesha kwamba ilianza St. Louis, Missouri.
St. Louis iko katika jimbo la Missouri, makao ya ardhi ya Missouri.
<answer>St. Louis</answer>

**Translated Output:**
The question asks where the Lewis and Clark expedition began.
The Lewis and Clark expedition was a journey across the land of North America to chart the course of the Missouri River and the early development of the United States.
It began in 1804 and involved Americans. Historical research and contextual analysis of the expedition indicate that it began in St. Louis, Missouri.
St. Louis is located in the state of Missouri, home to the Missouri Territory.
<answer>St. Louis</answer> |

Figure 12: **Generated outputs from a given question written in Swahili on GEOFACT-X.** All models use Swahili, but only M2A generates the correct answer, St Louis.

