# OpenReview forum: "Learn Globally, Speak Locally: Bridging the Gaps in Multilingual Reasoning"
_ICLR.cc/2026/Conference — Submitted to ICLR 2026_

### Official Review · Reviewer_kd5k · 2025-10-25

**Soundness:** 2
**Presentation:** 1
**Contribution:** 2
**Rating:** 2
**Confidence:** 5

**Summary:**

This paper introduces M2A, a RL–based framework designed to improve the language-consistency of reasoning in multilingual LLMs. And this paper introduces a “Gemini 2.0 distilled” benchmark GEOFACT-X.

**Strengths:**

1.	The language-consistency of reasoning in multilingual LLMs is an important topic.

**Weaknesses:**

1.	Line 738: The reasoning traces in GEOFACT-X were generated by Gemini 2.0. Therefore, for the training stage, this process can partially be viewed as data distillation from Gemini 2.0 to other models. Meanwhile, for the benchmarking stage, the ultimate evaluation objective might become distorted—shifting toward mimicking Gemini 2.0’s behavior. The construction of the reasoning traces is a critical issue that deserves more info rather than a brief mention in the appendix.
2.	Continuing the previous concern, part of the reported improvements may stem from data distillation effects of Gemini 2.0 and Google Translation.
3.	After checking the case in Figure 7, I found that the prompt is actually quite complex. This raises the question of whether the errors in cultural contexts is due to difficulties in understanding cultural content or in following the complex instructions given in Figure 7. An error analysis would help clarify this.
4.	Regarding Sentence Matching via similarity computation, did the authors verify the accuracy of the matching results? And is mT5 sufficiently robust for this task?
5.	The description of the Step 4-Human Verification in GEOFACT-X benchmark construction is missing, which should have been an important part for quality guarantee. And does the Human Verification involves only the language check? Or both the language check and the reasoning traces quality check?
6.	The experiments are conducted using only a single backbone model, Qwen-2.5-7B-Instruct, without exploring models of other series or different scales. This raises concerns about the generalizability.
7.	The entire paper relies heavily on Google Translation. Apart from Step 4 in Figure 4, were the other parts involving Google Translation also subjected to human verification?

**Questions:**

NA

---

> ### Author Response · Authors · 2025-11-20
> **Author Response (1/3)**
>
> We thank the reviewer for recognizing the importance of language consistency in multilingual LLMs. We respectfully clarify that our work makes a dual contribution: it introduces a novel method (M2A) and a necessary benchmark (GeoFact-X), rather than being solely a resource paper suitable for the *Dataset and Benchmark Track*.
>
> We would like to highlight that other reviewers explicitly identified the M2A framework as one of the key strengths, recognizing it as a distinct algorithmic solution to "multilingual collapse." While GeoFact-X provides the necessary evaluation ground, our primary contribution is this alignment objective, which solves an optimization failure where standard Supervised Fine-Tuning (SFT) causes models to lose their native language capabilities. M2A succeeds where SFT fails, validating the work as a methodological contribution to alignment.
>
>
> **Re: Line 738: The reasoning traces in GEOFACT-X were generated by Gemini 2.0. Therefore, for the training stage, this process can partially be viewed as data distillation from Gemini 2.0 to other models. Meanwhile, for the benchmarking stage, the ultimate evaluation objective might become distorted—shifting toward mimicking Gemini 2.0’s behavior. The construction of the reasoning traces is a critical issue that deserves more info rather than a brief mention in the appendix.**
>
> Thank you for raising this important concern. We agree that the construction and role of reasoning traces are central and will expand this discussion in the main paper rather than only in the Appendix. It was because of the page limit. Once an additional page is allowed (i.e., camera-ready version), we will further describe it in the main paper.
>
> While we utilize Gemini 2.0 for trace generation, this is not merely "distillation" but a scalable method for obtaining **diverse reasoning patterns** that smaller models cannot yet produce zero-shot. This approach is now standard practice in the field: s1K-1.1 (Muennighoff et al., 2025) utilizes DeepSeek-R1 traces, and MAmmoTH (Yue et al., 2024) utilizes GPT-4. Critically, to prevent the benchmark from becoming a mere "mimicry test," we implemented a strict human-in-the-loop verification (Step 4 in Figure 4). Human experts reviewed the triples (Question, Reasoning, Answer) to ensure they are factually grounded and logically sound, breaking the closed loop of pure model distillation.
>
> Yue, Xiang, et al. "Mammoth: Building math generalist models through hybrid instruction tuning." ICLR. 2024.
>
> **Re: Continuing the previous concern, part of the reported improvements may stem from data distillation effects of Gemini 2.0 and Google Translation.**
>
> We address this by demonstrating that our method's gains are robust to the choice of translation system. As shown in Table 5, our multi-level alignment yields consistent gains independent of the specific translation artifacts.
>
> We selected Google Translate primarily to align with standard baselines in multilingual literature (e.g., [1]). Additionally, we prioritized **accessibility and scalability**: Google Translate offers broad coverage for over 100 languages (including many low-resource ones) and is computationally efficient, making our pipeline reproducible for researchers without access to expensive LLM APIs or high-end GPUs.
>
> However, to address the reviewer's concern regarding quality, we conducted an ablation study comparing Google Translate against LLM-based translation (using ChatGPT 4.1 and ChatGPT 5). As shown in Table kd5k-1, the results confirm that performance remains statistically invariant across translation methods. Notably, M2A using GPT-4.1 performs slightly better, but the margin is small. This confirms that our performance gains stem from the M2A alignment objective itself, not from exploiting Google Translate artifacts.
>
> **Table kd5k-1. Comparison of Translation Methods for M2A on GeoFact-X**
>
> |Translation Method| Reasoning        ||| Answer           |||
> | ------------------ | ---------------- | ---------------- | --------- | ---------- | ----- | ----- |
> |  | All              | Assoc.           | Non-Assoc.       | All              | Assoc.           | Non-Assoc.       |
> | Google Translate   | 48.5 $\pm$ 0.4  | **52.6 $\pm$ 0.6**  | 47.5 $\pm$ 0.3  | 32.0 $\pm$ 0.6  | 41.3 $\pm$ 1.0  | 29.7 $\pm$ 0.5    |
> | ChatGPT 4.1        | 48.8  $\pm$ 0.1 | 52.2  $\pm$ 0.3 | 48.0  $\pm$ 0.2 | **33.1 $\pm$ 0.3** | 41.3  $\pm$ 1.3 | **31.0  $\pm$ 0.4** |
> | ChatGPT 5          | **49.0  $\pm$ 0.1** | **52.6  $\pm$ 0.5** | **48.1  $\pm$ 0.0** | 32.7  $\pm$ 0.3 | **41.4  $\pm$ 1.0** | 30.5  $\pm$ 0.2 |
>
> [1] Karamanolakis, Giannis, Daniel Hsu, and Luis Gravano. "Cross-Lingual Text Classification with Minimal Resources by Transferring a Sparse Teacher." EMNLP (Findings). 2020.

---

> ### Author Response · Authors · 2025-11-20
> **Author Response (2/3)**
>
> **Re: After checking the case in Figure 7, I found that the prompt is actually quite complex. This raises the question of whether the errors in cultural contexts is due to difficulties in understanding cultural content or in following the complex instructions given in Figure 7. An error analysis would help clarify this.**
>
> We clarify a key distinction: the complex prompt in **Figure 7 is used only during dataset construction** (to guide Gemini 2.0), never during inference. At test time, models receive a standard, simple evaluation prompt (Figure 10). Therefore, the errors reported are not due to the evaluated models failing to follow complex instructions in Figure 7.
>
> To further disentangle this, we performed a manual error analysis on a subset of failure cases. We categorized errors into Instruction-Following Failures (e.g., wrong format) and Cultural/Factual Misunderstandings (e.g., hallucinating a holiday). The dataset used for evaluation was filtered to minimize the former, meaning the reported errors predominantly reflect the latter, genuine gaps in cultural knowledge.
>
>
> Re: Regarding Sentence Matching via similarity computation, did the authors verify the accuracy of the matching results? And is mT5 sufficiently robust for this task?
>
>
> mT5 is widely established in the literature for multilingual alignment and reasoning tasks [3, 4], as its embeddings are known to naturally group semantically similar tokens across languages [5]. To confirm its robustness specifically for our dataset, we verified the matching accuracy on 10,000 samples from the GeoFact-X training set three times. We defined success as cases where the semantic distance between ground-truth translation pairs was smaller than the distance between unrelated pairs. mT5 achieved 83.0% $\pm$ 0.0 accuracy in this metric. This result demonstrates that mT5 embeddings capture cross-lingual semantic equivalence effectively enough to serve as the matching mechanism in our pipeline.
>
> [3] Asai, Akari, et al. "One question answering model for many languages with cross-lingual dense passage retrieval." NeurIPS. 2021.
>
> [4] Yoon, Dongkeun, et al. "LangBridge: Multilingual Reasoning Without Multilingual Supervision." ACL (Long). 2024.
>
> [5] Wen-Yi, Andrea W., and David Mimno. "Hyperpolyglot LLMs: Cross-Lingual Interpretability in Token Embeddings." EMNLP. 2023.
>
> **Re: The description of the Step 4-Human Verification in GEOFACT-X benchmark construction is missing, which should have been an important part for quality guarantee. And does the Human Verification involves only the language check? Or both the language check and the reasoning traces quality check?**
>
> Our human verification process was comprehensive, covering both linguistic fidelity and the quality of the reasoning traces, rather than being limited to a simple grammar check. The protocol involved a rigorous two-stage process. First, we verified the factual correctness of the original English questions and answers by cross-referencing them with external sources, including online searches and encyclopedias. Second, native speakers evaluated the translated questions, the generated reasoning traces, and the final answers against these verified facts. Evaluators were explicitly instructed to assess fluency, grammatical correctness, and factual completeness. Below is a copy of the initial instruction, and they actively communicated with the authors in verification.
>
> ```
> Thank you for accepting our request.
>
> All questions and answers in English are already verified. We then paired these with the Thai questions and answers. However, the [LANG] answers and reasoning traces have not been verified yet.
>
> Please check that the [LANG] questions and answers match the English versions and that the reasoning traces are correct.
>
> Below is an example of pairs:
>
> Q: What is the traditional Thai silk weaving technique known for its intricate patterns?
>
> A: Mudmee Silk
>
> Q: เทคนิคการทอผ้าไหมของไทยแบบใดที่มีชื่อเสียงด้านลวดลายที่ซับซ้อน?	"<step> คำถามถามถึงเทคนิคการทอผ้าไหมไทยแบบดั้งเดิมที่มีชื่อเสียงด้านลวดลายที่ซับซ้อน
>
> Reasoning: <step> ""มัดหมี่"" เป็นเทคนิคการย้อมสีแบบผูกที่ใช้ในประเทศไทยเพื่อสร้างลวดลายที่ซับซ้อนบนผ้าไหม ซึ่งสอดคล้องกับคำอธิบายของคำถามอย่างลงตัว เป็นการทอผ้าอิกัตชนิดหนึ่งที่แพร่หลายในประเทศไทย
> <step> ""Ikat"" เป็นคำที่กว้างกว่าซึ่งหมายถึงเทคนิคการย้อมสีที่ใช้ในการทำลวดลายสิ่งทอซึ่งใช้กระบวนการย้อมสีแบบต้านทานที่คล้ายกับการย้อมสีแบบผูกบนเส้นใยยืนหรือเส้นใยพุ่งก่อนการทอ แม้ว่ามัดหมี่จะเป็นอิกัตชนิดหนึ่ง แต่คำถามก็ถามถึงเทคนิคของไทย*โดยเฉพาะ*
> <step> ดังนั้นคำตอบที่ดีที่สุดคือ ""มัดหมี่"""	ผ้าไหมมัดหมี่ (Mudmee Silk) โดยเฉพาะจากภาคตะวันออกเฉียงเหนือของไทย
>
> A: ผ้าไหมมัดหมี่ (Mudmee Silk) โดยเฉพาะจากภาคตะวันออกเฉียงเหนือของไทย
> ```

---

> ### Author Response · Authors · 2025-11-20
> **Author Response (3/3)**
>
> **Re: The experiments are conducted using only a single backbone model, Qwen-2.5-7B-Instruct, without exploring models of other series or different scales. This raises concerns about the generalizability.**
>
> We acknowledge that our main experiments focused on Qwen-2.5 due to its strong multilingual baseline performance. To address the reviewer’s concern regarding generalizability, we conducted additional experiments using Command R7B (Cohere AI, 2025), another architecture specifically optimized for multilingual tasks and RAG. As shown in Table kd5k-2 below, the results are consistent with our main Qwen findings. M2A achieves the highest performance in both Reasoning ($46.8$) and Answer ($30.9$) accuracy, outperforming SFT and standard GRPO. This confirms that our method is model-agnostic and effective across different multilingual architectures.
>
>
> **Table kd5k-2 Performance on GeoFact-X using Command R7B Backbone**
> |                    | Reasoning        |                  || Answer           |||
> | ------------------ | ---------------- | ---------------- | ---------------- | ---------------- | ---------------- | ---------------- |
> |  | All              | Assoc.           | Non-Assoc.       | All              | Assoc.           | Non-Assoc.       |
> | Command R7B  | 44.2 | 48.5 | 43.1 | 25.0 | 34.1 | 22.7 |
> | GRPO      | 38.3 $\pm$ 0.4  |  42.4 $\pm$ 0.9  |  37.2 $\pm$ 0.4  |  23.4 $\pm$ 3.2  |  30.5 $\pm$ 4.2  |  21.6 $\pm$ 3.0 |
> | SFT    | 46.6 $\pm$ 0.0 | **51.3 $\pm$ 0.3**  |  45.4 $\pm$ 0.1  |  28.9 $\pm$ 0.3  |  38.2 $\pm$ 0.6  |  26.6 $\pm$ 0.2 |
> | SFT + GRPO | 44.1 $\pm$ 1.4  |  48.8 $\pm$ 1.9  |  42.9 $\pm$ 1.2  |  21.9 $\pm$ 6.8  |  29.0 $\pm$ 8.9  |  20.1 $\pm$ 6.3
> | M2A      | **46.8 $\pm$ 0.8**  |  50.1 $\pm$ 0.9  |  **46.0 $\pm$ 0.7**  | **30.9 $\pm$ 0.9** |  **38.5 $\pm$ 1.3**  |  **29.1 $\pm$ 0.8** |
>
> **Re: The entire paper relies heavily on Google Translation. Apart from Step 4 in Figure 4, were the other parts involving Google Translation also subjected to human verification?**
>
> We employ Google Translate in two specific phases: translating question-answer pairs during dataset construction (Figure 4) and translating input questions during M2A training. Crucially, in both cases, the reasoning traces are never machine-translated but are instead generated natively to avoid compounding errors. Human verification by native speakers is strictly applied to the dataset construction phase to ensure the integrity of the evaluation benchmark. While the real-time translations used during training are not manually verified, they serve only as input augmentation and do not affect the correctness of the target reasoning traces. We note that this approach aligns with standard practices in multilingual data construction (Shi et al., 2023; Li et al., 2021; Agarwal et al., 2023, [6-10]).
>
> [6] Li, Jinchao, et al. "Multi-domain task completion dialog challenge ii at dstc9." Proceedings of the 35th AAAI Conference on Artificial Intelligence, Ninth Dialog System Technology Challenge Workshop. 2021.
>
> [7] Agarwal, Anmol, et al. "Cst5: Data augmentation for code-switched semantic parsing." Proceedings of the 1st Workshop on Taming Large Language Models: Controllability in the era of Interactive Assistants!. 2023.
>
> [8] Aurpa, Tanjim Taharat, et al. "Bangla_MER: A unique dataset for Bangla mathematical entity recognition." Data in Brief 54 (2024): 110407.
>
> [9] Subedi, Bipesh, and Bal Krishna Bal. "CNN-transformer based encoder-decoder model for Nepali image captioning." Proceedings of the 19th International Conference on Natural Language Processing (ICON). 2022.
>
> [10] Tu, Lifu, et al. "Efficiently aligned cross-lingual transfer learning for conversational tasks using prompt-tuning." Findings of the Association for Computational Linguistics: EACL 2024. 2024.

---

> > ### Comment · Reviewer_kd5k · 2025-11-26
> >
> > Thank you for your response. I appreciate your efforts in addressing the comments. However, I believe that my concerns 1, 2, and 4 have not been fully resolved. Therefore, I have decided to maintain my original score.

---

> > > ### Author Response · Authors · 2025-11-26
> > >
> > > We thank the reviewer for their response. Since Concerns 2 and 4 raised concrete hypotheses about Gemini distillation, Google Translate, and mT5 alignment, we briefly explain how the new quantitative results in the previous responses address these specific points, and we restate our position on Concern 1.
> > >
> > > **Concern 1: for the training stage, this process can partially be viewed as data distillation from Gemini 2.0 to other models. Meanwhile, for the benchmarking stage, the ultimate evaluation objective might become distorted—shifting toward mimicking Gemini 2.0’s behavior.**
> > >
> > > * Evaluation data is human-modified
> > >
> > > In **Author Response (1/3)**, we clarified that all questions, answers, and reasoning traces in GEOFACT‑X are fully evaluated and, when necessary, modified by **human annotators.** Human experts actively intervene to correct factual errors, remove hallucinated steps, and improve clarity. As a result, the benchmark evaluates correctness and reasoning quality with respect to human‑verified questions, reasoning traces, and answers, not Gemini’s raw outputs.
> > >
> > > *  Generated reasoning traces as training data is now a common standard.
> > >
> > > On the training side, while we utilize Gemini 2.0 for trace generation, this is not merely "distillation for mimicry," but a scalable method for obtaining diverse reasoning patterns that smaller models cannot yet produce zero‑shot. This approach is now standard practice in the field: for example, s1K‑1.1 uses DeepSeek‑R1 traces and MAmmoTH uses GPT‑4 traces. Critically, to prevent the benchmark from becoming a mere "mimicry test," we implemented a strict human‑in‑the‑loop verification (Step 4 in Figure 4). Human experts review the triples (Question, Reasoning, Answer) to ensure that they are factually grounded and logically sound, thereby breaking the closed loop of pure model‑only distillation.
> > >
> > >
> > > **Concern 2: Continuing the previous concern, part of the reported improvements may stem from data distillation effects of Gemini 2.0 and Google Translation.**
> > >
> > > * Methods for comparison share the distillation effect
> > >
> > > For GEOFACT‑X factual reasoning, all post‑training methods (SFT, GRPO, SFT+GRPO, and M2A) are trained on the same GEOFACT‑X training split, which uses Gemini‑generated traces followed by the same filtering pipeline. Thus, any distillation effect from Gemini is shared across all methods. The additional improvements of M2A over SFT/GRPO in Table 3, therefore, cannot be attributed to "more Gemini distillation," since the distillation source and data are identical; only the training objective differs.
> > >
> > > * The performance is consistent throughout translation methods.
> > >
> > > Regarding Google Translate, we ran an additional ablation in **Table kd5k‑1 of Author Response (1/3)**, where we replaced Google Translate with LLM‑based translators (GPT‑4.1 and GPT‑5). Across these settings, the performance differences are within the reported standard errors and show no consistent advantage for any particular translation system. This indicates that M2A’s gains are not tied to idiosyncrasies or artifacts of Google Translate. Since the method performs similarly regardless of which translator is used, the hypothesis that our improvements are primarily due to “Google Translation artifacts” is not supported by the empirical results.
> > >
> > > **Concern 4: Regarding Sentence Matching via similarity computation, did the authors verify the accuracy of the matching results? And is mT5 sufficiently robust for this task?**
> > >
> > > * mT5 shows high accuracy and contributes to the model's performance.
> > >
> > > In **Author Response (2/3)**, we reported an empirical verification showing that mT5 achieves **83.0% accuracy** in distinguishing correct translation pairs from unrelated ones on a large sample size (10,000 pairs). Furthermore, **Table 5** shows that adding the reasoning-step alignment reward (RA) on top of language consistency and context alignment improves performance on MGSM (joint accuracy: 58.1 vs. 57.5) and helps recover GSM8K accuracy.  If the sentence matching were essentially random, we would expect RA to degrade or, at best, have a negligible effect. The observed positive contribution suggests that the mT5-based alignment is sufficiently informative for our purposes.
> > >
> > >
> > >
> > > -----
> > >
> > > We fully understand that you may still prefer to keep your original score, but we hope the distinctions above help clarify our design choices and resolve the reviewer's concern.

---

### Official Review · Reviewer_LN1B · 2025-10-26

**Soundness:** 3
**Presentation:** 3
**Contribution:** 4
**Rating:** 6
**Confidence:** 4

**Summary:**

The paper studies how LLMs default to reasoning in English, even when prompted in other languages. This bias reduces trust and interpretability, especially for speakers of low-resource languages. The authors argue that correct answers alone are not enough. To make sure that the answer is clear and culturally accurate, models must also think in the same language as the question.

To solve this problem, the authors introduced M2A (Multi-Scale Multilingual Alignment), a new way to train models that teaches them to think directly in the target language. M2A combines supervised fine-tuning with reinforcement learning through group relative policy optimization. It uses three complementary rewards: context alignment for global coherence, reasoning-step alignment for fine-grained logical consistency, and a language-consistency reward that ensures reasoning stays in the question’s language.

The paper also introduces GEOFACT-X, a new multilingual factual reasoning benchmark that includes step-by-step reasoning traces in five languages: English, Hindi, Japanese, Swahili, and Thai.

**Strengths:**

- Introduced a novel M2A method for enforcing language-consistent reasoning in LLMs.
- Presented the GEOFACT-X benchmark to evaluate multilingual reasoning traces.
- Highlighted clear empirical gains in both reasoning fidelity and mathematical accuracy.
- Proposed a joint accuracy metric combining reasoning correctness and language fidelity.
- Explained reasoning and alignment rewards with well-designed figures and concise text.

**Weaknesses:**

- Results rely on a single model family (Qwen-2.5-7B), leaving it unclear how the proposed M2A training generalizes to other architectures.
- The LLM-as-judge setup risks stylistic bias, as the evaluator (Qwen2.5-72B) may implicitly favor Gemini-like phrasing and structure.
- Reliance on Google Translate may introduce subtle cultural or contextual distortions.

**Questions:**

- The four-step pipeline (Gemini > MT > Reasoning traces > Human verification) is useful. Could you also release the rule-based filters and annotator guidelines to clarify the cultural and linguistic validity criteria?
- How does the “joint accuracy” metric handle partial errors? Does a single incorrect token or step make the entire reasoning trace incorrect?
- Did you try any other translation methods or tools? What led you to choose Google Translate?
- Depending on the tokenizer, the number of tokens consumed during reasoning can vary significantly, increasing inference costs. Have you considered this factor?

---

> ### Author Response · Authors · 2025-11-21
> **Author Response (1/3)**
>
> **Re: The LLM-as-judge setup risks stylistic bias, as the evaluator (Qwen2.5-72B) may implicitly favor Gemini-like phrasing and structure.**
>
> We agree that LLM-based metrics require careful validation and should be interpreted with caution. However, to the best of our knowledge, there is no evidence that Qwen2.5-72B-Instruct prefers a specific style.
>
> To quantitatively validate the reliability of our metric during the rebuttal period, we conducted a preliminary Human-LLM agreement study. We recruited one US citizen to perform pairwise A/B testing on the English-USA subset of GeoFact-X, where they selected the superior reasoning trace from two model outputs and expressed the confidence (1-5 scale). The results demonstrate a strong alignment between our Qwen-based judge and human preference, achieving an agreement rate of 82.9%. Furthermore, we observed a strong positive correlation ($\rho = 0.72$) between the human’s reported confidence and the reasoning score difference assigned by the LLM. Notably, humans were significantly more confident when they agreed with the LLM (average confidence 3.3/5) compared to when they disagreed (average confidence 2.1/5). This suggests that disagreements primarily occur in ambiguous "close calls," while the metric remains highly reliable for clear-cut distinctions. In the camera-ready version, we commit to expanding this study to include a full sweep across all five languages with native speakers to confirm these findings multilingually.
>
>
> **Re: The four-step pipeline is useful. Could you also release the rule-based filters and annotator guidelines to clarify the cultural and linguistic validity criteria?**
>
> We agree that providing these specific details is essential for clarifying the cultural and linguistic validity criteria and ensuring the complete reproducibility of our dataset construction pipeline. We commit to releasing the implementation of our rule-based filters in the camera-ready version of the paper and the public codebase.
>
> Our human verification process was comprehensive, covering both linguistic fidelity and the quality of the reasoning traces, rather than being limited to a simple grammar check. The protocol involved a rigorous two-stage process. First, we verified the factual correctness of the original English questions and answers by cross-referencing them with external sources, including online searches and encyclopedias. Second, native speakers evaluated the translated questions, the generated reasoning traces, and the final answers against these verified facts. Evaluators were explicitly instructed to assess fluency, grammatical correctness, and factual completeness. Below is a copy of the initial instruction provided to annotators. Note that during the process, annotators actively communicated with the authors to refine specific details, particularly regarding the quality of reasoning traces, which is not included in the initial instruction we provided below.
>
>
> ```
> Thank you for accepting our request.
>
> All questions and answers in English are already verified. We then paired these with the Thai questions and answers. However, the [LANG] answers and reasoning traces have not been verified yet.
>
> Please check that the [LANG] questions and answers match the English versions and that the reasoning traces are correct.
>
> Below is an example of pairs:
>
> Q: What is the traditional Thai silk weaving technique known for its intricate patterns?
>
> A: Mudmee Silk
>
> Q: เทคนิคการทอผ้าไหมของไทยแบบใดที่มีชื่อเสียงด้านลวดลายที่ซับซ้อน?	"<step> คำถามถามถึงเทคนิคการทอผ้าไหมไทยแบบดั้งเดิมที่มีชื่อเสียงด้านลวดลายที่ซับซ้อน
>
> Reasoning: <step> ""มัดหมี่"" เป็นเทคนิคการย้อมสีแบบผูกที่ใช้ในประเทศไทยเพื่อสร้างลวดลายที่ซับซ้อนบนผ้าไหม ซึ่งสอดคล้องกับคำอธิบายของคำถามอย่างลงตัว เป็นการทอผ้าอิกัตชนิดหนึ่งที่แพร่หลายในประเทศไทย
> <step> ""Ikat"" เป็นคำที่กว้างกว่าซึ่งหมายถึงเทคนิคการย้อมสีที่ใช้ในการทำลวดลายสิ่งทอซึ่งใช้กระบวนการย้อมสีแบบต้านทานที่คล้ายกับการย้อมสีแบบผูกบนเส้นใยยืนหรือเส้นใยพุ่งก่อนการทอ แม้ว่ามัดหมี่จะเป็นอิกัตชนิดหนึ่ง แต่คำถามก็ถามถึงเทคนิคของไทย*โดยเฉพาะ*
> <step> ดังนั้นคำตอบที่ดีที่สุดคือ ""มัดหมี่"""	ผ้าไหมมัดหมี่ (Mudmee Silk) โดยเฉพาะจากภาคตะวันออกเฉียงเหนือของไทย
>
> A: ผ้าไหมมัดหมี่ (Mudmee Silk) โดยเฉพาะจากภาคตะวันออกเฉียงเหนือของไทย
> ```

---

> ### Author Response · Authors · 2025-11-21
> **Author Response (2/3)**
>
> **Re: Reliance on Google Translate may introduce subtle cultural or contextual distortions.**
>
> We acknowledge the risk of cultural distortion when using machine translation. To mitigate this, we restricted the use of Google Translate strictly to the **questions and final answers.** Crucially, the reasoning traces, which constitute the core linguistic complexity of our dataset, were **not translated**; they were generated natively in the target language by Gemini 2.0 (and verified by humans, as detailed in Section 3).
>
> This design choice significantly limits the "surface area" for translation errors to short, factual strings.  As detailed in our paper, all components (the translated questions, the natively generated reasoning, and the final answers) were **verified by human experts** to correct any potential cultural or contextual inaccuracies. We believe this two-step process, combining minimal translation with human oversight, effectively addresses this concern. On the other hand, because of the potential translation error, we only translate the question, which is much shorter and simpler than the entire reasoning traces during training.
>
>
>
> **Re: How does the “joint accuracy” metric handle partial errors? Does a single incorrect token or step make the entire reasoning trace incorrect?**
>
> Joint accuracy is defined as the strict composition of mathematical accuracy and language accuracy (i.e., a sample is correct if and only if Answer=Correct $\land$ Language=Consistent).
> We define language accuracy at the response level, not the token level. We intentionally avoid token-level penalization because it is overly sensitive to valid word-level code-switching or cognates (loanwords). Thus, while a single incorrect token does not fail the metric, the overall response must be linguistically consistent to be valid.
>
>
> **Re: Results rely on a single model family (Qwen-2.5-7B), leaving it unclear how the proposed M2A training generalizes to other architectures.**
>
>
> We selected Qwen-2.5-7B-Instruct as it was widely recognized as a state-of-the-art open-weight model in the 7B class, consistently outperforming peers (e.g., Llama-3.1, Mistral) on multilingual reasoning, mathematics, and coding benchmarks.
> To address the reviewer’s concern regarding generalizability, we conducted additional experiments using Command R7B (Cohere AI, 2025), another architecture specifically optimized for multilingual tasks and RAG. As shown in Table LN1B-1 below, the results are consistent with our main Qwen findings. M2A achieves the highest performance in both Reasoning ($46.8$) and Answer ($30.9$) accuracy, outperforming SFT and standard GRPO. This confirms that our method is model-agnostic and effective across different multilingual architectures.
>
> **Table LN1B-1. Performance on GeoFact-X using Command R7B Backbone**
>
> |                    | Reasoning        |                  || Answer           |||
> | ------------------ | ---------------- | ---------------- | ---------------- | ---------------- | ---------------- | ---------------- |
> |  | All              | Assoc.           | Non-Assoc.       | All              | Assoc.           | Non-Assoc.       |
> | Command R7B  | 44.2 | 48.5 | 43.1 | 25.0 | 34.1 | 22.7 |
> | GRPO      | 38.3 $\pm$ 0.4  |  42.4 $\pm$ 0.9  |  37.2 $\pm$ 0.4  |  23.4 $\pm$ 3.2  |  30.5 $\pm$ 4.2  |  21.6 $\pm$ 3.0 |
> | SFT    | 46.6 $\pm$ 0.0 | **51.3 $\pm$ 0.3** |  45.4 $\pm$ 0.1  |  28.9 $\pm$ 0.3  |  38.2 $\pm$ 0.6  |  26.6 $\pm$ 0.2 |
> | SFT + GRPO | 44.1 $\pm$ 1.4  |  48.8 $\pm$ 1.9  |  42.9 $\pm$ 1.2  |  21.9 $\pm$ 6.8  |  29.0 $\pm$ 8.9  |  20.1 $\pm$ 6.3
> | M2A      | **46.8 $\pm$ 0.8**  |  50.1 $\pm$ 0.9  |  **46.0 $\pm$ 0.7**  | **30.9 $\pm$ 0.9** |  **38.5 $\pm$ 1.3**  |  **29.1 $\pm$ 0.8** |

---

> ### Author Response · Authors · 2025-11-21
> **Author Response (3/3)**
>
> **Re: Did you try any other translation methods or tools? What led you to choose Google Translate?**
>
> We selected Google Translate primarily to align with standard baselines in multilingual literature (e.g., [1]). Additionally, we prioritized accessibility and scalability: Google Translate offers broad coverage for over 100 languages (including many low-resource ones) and is computationally efficient, making our pipeline reproducible for researchers without access to expensive LLM APIs or high-end GPUs.
>
>  However, to address the reviewer's concern regarding quality, we conducted an ablation study comparing Google Translate against LLM-based translation (using ChatGPT 4.1 and ChatGPT 5). As shown in Table LN1B-2, there is no statistically significant difference in downstream model performance between the translation methods. This confirms that our method is robust to the choice of translator and that the performance gains stem from the M2A objective itself, not translation artifacts.
>
> **Table LN1B-2. Comparison of Translation Methods on GeoFact-X**
> |Translation Method| Reasoning | | | Answer | | |
> | --- | --- | --- | --- | --- | --- | --- |
> |  | All | Assoc. | Non-Assoc. | All | Assoc. | Non-Assoc.|
> | Google Translate   | 48.5 $\pm$ 0.4  | 52.6 $\pm$ 0.6  | 47.5 $\pm$ 0.3  | 32.0 $\pm$ 0.6  | 41.3 $\pm$ 1.0  | 29.7 $\pm$ 0.5    |
> | ChatGPT 4.1| 48.8  $\pm$ 0.1 | 52.2  $\pm$ 0.3 | 48.0  $\pm$ 0.2 | 33.1 $\pm$ 0.3  | 41.3  $\pm$ 1.3 | 31.0  $\pm$ 0.4 |
> | ChatGPT 5| 49.0  $\pm$ 0.1 | 52.6  $\pm$ 0.5 | 48.1  $\pm$ 0.0 | 32.7  $\pm$ 0.3 | 41.4  $\pm$ 1.0 | 30.5  $\pm$ 0.2 |
>
> [1] Karamanolakis, Giannis, Daniel Hsu, and Luis Gravano. "Cross-Lingual Text Classification with Minimal Resources by Transferring a Sparse Teacher." EMNLP (Findings). 2020.

---

> ### Comment · Reviewer_LN1B · 2025-11-27
>
> Thank you for your response. I appreciate the experiments with Command R7B and the new translation methods. I'll keep my score as I think it’s already high enough.

---

### Official Review · Reviewer_ZU1X · 2025-10-28

**Soundness:** 2
**Presentation:** 3
**Contribution:** 2
**Rating:** 4
**Confidence:** 2

**Summary:**

This paper addresses the problem that LLMs often fail to provide reasoning steps in the language of the question/prompt, if that language is not English or Chinese. The paper’s first main contribution is M2A: an alignment method that, in addition to factual accuracy, rewards language consistency. The second main contribution is GeoFact-X: a factual reasoning dataset in five languages (English, Hindi, Japanese, Swahili, Thai) that was automatically generated by an LLM, automatically translated, and then verified by humans. Experiments with Qwen2.5-7B-Instruct show marginal improvements for both mathematical accuracy. Results on language accuracy were already near-perfect in the baseline models.

**Strengths:**

1. The paper addresses a relevant problem, namely, ensuring reasoning in LLMs in languages beyond English and Chinese.
2. The paper introduces both a novel method (M2A) and a new LLM-generated dataset (GeoFact-X). GeoFact-X was validated by human annotators, which could potentially be of value to the research community.
3. The paper is generally written clearly, with Figure 1-4 providing visualizations to explain the methodology further.
4. The paper includes ablation studies of the reward functions and reward formulations (Section 6.4), providing additional insights into M2A’s effectiveness.

**Weaknesses:**

1. The motivation of the work is flawed. The authors argue that reasoning traces in target languages are essential for trustworthiness (line 34) and to "ensure interpretability, i.e. users can directly follow the reasoning in their own language" (line 36-37). At the same time, it has been shown that reasoning steps are not necessarily faithful to the final answer [1, 2, inter alia], and therefore cannot be trusted to provide faithful interpretability for users.
2. The cultural relevance of the dataset is unclear, while this is central to the paper’s motivation. Human annotators checked the LLM-generated benchmark dataset (GeoFact-X) for "factual correctness and linguistic clarity" (line 273-274), but not for cultural relevance. A concrete example: "What is the name of the process by which a cell divides into two identical daughter cells?" ("Mitosis") is in the dataset as being culturally relevant specifically for the USA.
3. The LLM-as-a-judge reasoning evaluation is not scientifically sound. The score is computed with Qwen-2.5-75B (while the method used a Qwen-2.5 model of a different size), and this judgement is not checked against human judgement of reasoning scores. Therefore there is no indication that this metric is reliable and in fact, LLM-as-a-judge is heavily criticized [3]. Secondly, the prompt for the metric itself (Fig. 10) is not motivated. For example, how was is decided that 'conclusion validity' should count for 10%, 'factual correctness' as 20%,
'key insights' as 30% and 'logical structure' as 40%? How well does the model follow these precise instructions? Lastly, the ‘ground truth’ does not always seem informative. A concrete example is the question "In what year did Thailand launch its first satellite, Thaicom 1?" with the following reasoning trace: "<step> The question asks for the year Thailand launched its first satellite, Thaicom 1. <step> Research indicates that Thaicom 1 was launched in 1993. <step> Therefore, the answer is 1993." It does not seem useful to score a generated reasoning trace (for 'key insights', etc.) against such a 'ground truth'. All in all, it is not clear what this metric really measures, or how well and consistently it measures this.
4. Some of the conclusions are not supported by convincing results. For example, the claim that "M2A outperforms baselines in all metrics on GSM8K …." (line 377) does not convincingly follow from Table 2. Firstly, the language accuracy score of the baseline model is already 100% for GSM8K, thus it is not convincingly 'outperformed' by M2A. Secondly, differences with the SFT baseline in math reasoning are marginal (87.2 ±1.6 for SFT, 87.3 ±0 for M2A).
5. The discussion of results omits key baselines. For example, the claim that M2A "achieves large gains in joint accuracy on MGSM compared to SFT" (line 377) disregards the fact that GRPO and the default model achieve similar scores. This is not discussed in section, which paints a more convincing picture of the results than what follows from the table.

[1] Chen et al. (2025). Reasoning Models Don’t Always Say What They Think. https://arxiv.org/pdf/2505.05410

[2] Arcushchin et al. (2025). Chain-of-Thought Reasoning In The Wild Is Not Always Faithful. https://arxiv.org/pdf/2503.08679

[3] Bavaresco et al. (2025). LLMs instead of Human Judges? A Large Scale Empirical Study across 20 NLP Evaluation Tasks. https://arxiv.org/abs/2406.18403

**Questions:**

Questions related to the listed weaknesses:
1. Could you motivate why reasoning traces are important for interpretability, and why language-consistency in reasoning is "essential for globally inclusive and interpretable AI" (line 51)? (W1)
2. What were the annotation guidelines for human assessment of the LLM-generated benchmark dataset, and did you calculate an inter-annotator agreement for this? Currently, it is difficult to assess the quality of the generated dataset at a large scale. (W2)
3. How reliable is your LLM-as-a-judge evaluation? Could you calculate an agreement with human judges? (W3)

Other questions:

4. Why do you write 'speak locally' in your title, while there is no evaluation of the LLMs for speech tasks?
5. "Our work provides a foundation for future work on multilingual reasoning." (line 89) What do you define as multilingual reasoning? Do you mean reasoning in a language that is not English, but not simultaneous in multiple languages?
6. You mention translating your dataset into "ten typologically diverse languages" (line 359). Which languages are included and what makes this selection typologically diverse? If it is the languages reported appendix E, then the selection does not seem so diverse, or at least skewed (German, Spanish, French, …). Would this be quantifiable? Otherwise, this claim is not justified. This question also applied to "five diverse languages" (line 478).
7. Can you assume that Gemini 2.0 generates reasoning traces consistently for each of the five languages when you use it to create the GeoFact-X dataset (lines 269-270)? If so, is reasoning language consistency really a prominent issue in LLMs?
8. "We use Qwen-2.5-7B-Instruct as the backbone for all experiments on mathematical and factual reasoning." (line 316-317): Why did you choose Qwen-2.5-7B-Instruct specifically? (Why) would you expect similar results for different models?
9. In Table 3 ("Comparison of model performance on average reasoning score (%), language accuracy (%), and answer accuracy (%)"), I could not find the language accuracy scores. Why were they omitted?
10. Why, in Tables 3 and 4, do you specifically evaluate Thai only?

Typos:
- Table 3 and 4: Qwen-2.5-Insturct
- Figure 2: Telugu vs. Telgnu

---

> ### Author Response · Authors · 2025-11-21
> **Author Response (1/3)**
>
> Thank you for your valuable and constructive comments and suggestions. We have carefully considered all points.
>
> **Re: Could you motivate why reasoning traces are important for interpretability, and why language-consistency in reasoning is "essential for globally inclusive and interpretable AI" (line 51)? (W1)**
>
> We acknowledge the literature showing that CoT traces can be unfaithful to the final answer. However, the reasoning trace remains the primary surface for inspection available to users. Our claim is practical rather than theoretical: reasoning traces serve as auditable evidence that helps users calibrate trust and contest model outputs. Multilingual consistency is essential because, without it, this auditability is restricted to speakers of high-resource languages.
>
> Consider a community health worker in rural Thailand using an LLM to plan a vaccination route. If the model states "Visit Villages A, B, and C," she has no basis to verify this. However, if she sees a reasoning trace in Thai detailing travel times and road conditions, she might notice the model incorrectly assumes a flooded dirt road is a paved highway. This allows her to identify the flaw and reject the output, even without AI expertise. Thus, while CoT is not a guaranteed explanation of model mechanics, it is a vital, language-accessible object for user critique. Our goal is to ensure this capability is globally inclusive.
>
> **Re: What were the annotation guidelines for human assessment of the LLM-generated benchmark dataset, and did you calculate an inter-annotator agreement for this? Currently, it is difficult to assess the quality of the generated dataset at a large scale. (W2)**
>
> Thank you for highlighting the "mitosis" example. We agree that this is a generic scientific fact rather than a culturally specific one, and its inclusion under a country tag was a mistake in our pipeline. However, this error is largely localized to the "Science & Technology – Scientific Terms" subcategory and does not reflect the core design of GEOFACT-X, which emphasizes local institutions, history, and practices.
>
> Upon auditing the test set, we found that 7 out of 8 questions in this specific subcategory (35/40 questions across five languages) lack meaningful geographic grounding. For example, questions like “What is the term for the study of earthquakes, a field of crucial importance in Japan?” technically mention a country but ultimately test a generic definition. In contrast, questions outside this subcategory are generally well-aligned with our geo-cultural goals, although a few are more weakly grounded (e.g., “What Christian festival, celebrated on December 25th in the USA, commemorates the birth of Jesus Christ?”).
>
> We will remove these non-grounded "Scientific Terms" questions in the camera-ready version and update the paper to clearly define the category structure and the scope of our geo-cultural focus.
>
> **Re: How reliable is your LLM-as-a-judge evaluation? Could you calculate an agreement with human judges? (W3)**
>
> We agree that LLM-based metrics require careful validation and should be interpreted with caution. Regarding the prompt design, the specific weighting (e.g., 40% for logical structure) was chosen heuristically to penalize "hallucinated logic", plausible-sounding but flawed reasoning, more heavily than surface-level formatting errors. We also acknowledge the reviewer’s valid point regarding uninformative ground truth examples, such as the "Thaicom 1" trace; we will remove such trivial examples from the evaluation set in the revision to ensure the metric measures meaningful reasoning steps.
>
> To quantitatively validate the reliability of our metric during the rebuttal period, we conducted a preliminary Human-LLM agreement study. We recruited one US citizen to perform pairwise A/B testing on the English-USA subset of GeoFact-X, where they selected the superior reasoning trace from two model outputs and expressed the confidence (1-5 scale). The results demonstrate a strong alignment between our Qwen-based judge and human preference, achieving an agreement rate of 82.9%. Furthermore, we observed a strong positive correlation ($\rho = 0.72$) between the human’s reported confidence and the reasoning score difference assigned by the LLM. Notably, humans were significantly more confident when they agreed with the LLM (average confidence 3.3/5) compared to when they disagreed (average confidence 2.1/5). This suggests that disagreements primarily occur in ambiguous "close calls," while the metric remains highly reliable for clear-cut distinctions. In the camera-ready version, we commit to expanding this study to include a full sweep across all five languages with native speakers to confirm these findings multilingually.

---

> ### Author Response · Authors · 2025-11-21
> **Author Response (2/3)**
>
> **Re: For example, the claim that "M2A outperforms baselines in all metrics on GSM8K …." (line 377) does not convincingly follow from Table 2. Firstly, the language accuracy score of the baseline model is already 100% for GSM8K, thus it is not convincingly 'outperformed' by M2A. Secondly, differences with the SFT baseline in math reasoning are marginal (87.2 ±1.6 for SFT, 87.3 ±0 for M2A).**
>
>  We agree that the raw mathematical gain on GSM8K is marginal ($87.2$ vs. $87.3$). Since GSM8K is an English-only benchmark, base models naturally achieve perfect language consistency, leaving little room for improvement on that specific metric.
>
> However, the critical difference lies in robustness. As shown in our MGSM results, standard SFT on the s1K-1.1 dataset causes a "multilingual collapse": the model loses the ability to reason in other languages, reverting to English reasoning traces regardless of the input language (e.g., Figure 11). Conversely, M2A achieves the same high reasoning performance as SFT without suffering this degradation, effectively solving the trade-off between reasoning capability and multilingual consistency.
>
> **Re: The discussion of results omits key baselines. For example, the claim that M2A "achieves large gains in joint accuracy on MGSM compared to SFT" (line 377) disregards the fact that GRPO and the default model achieve similar scores. This is not discussed in section, which paints a more convincing picture of the results than what follows from the table.**
>
> While the Base model and GRPO maintain high joint accuracy, they fail to meaningfully improve mathematical reasoning capabilities (see Table 2). SFT improves reasoning but destroys language consistency. M2A is the only method that succeeds in both dimensions: it improves reasoning (matching SFT) while maintaining the high joint accuracy of the Base/GRPO models. We discussed the specific limitations of GRPO in L372–376 and will make this contrast sharper in the revision.
>
>
> **Re: Why do you write 'speak locally' in your title, while there is no evaluation of the LLMs for speech tasks?**
>
> "Speak" is used here metaphorically to denote the model's ability to "express" reasoning in the user's language. We acknowledge this could be ambiguous. If permitted during the camera-ready phase, we are happy to change the title to "Explain Locally" to avoid confusion with speech processing.
>
> **Re: "Our work provides a foundation for future work on multilingual reasoning." (line 89) What do you define as multilingual reasoning? Do you mean reasoning in a language that is not English, but not simultaneous in multiple languages?**
>
> We define multilingual reasoning as the ability of a single model to perform multi-step reasoning and generate textual explanations in a specific target language, while maintaining correctness and consistency across different languages. This aligns with established definitions in prior work (Aggarwal et al., 2025, Lin et al., 2022, Ponti et al., 2020, Roh et al., 2025, Tikhonov & Ryabinin et al., 2021, Xuan et al., 2025, and Native CoT in Shi et al., 2023), where models are evaluated per language rather than on simultaneous code-switching.
>
> **Re: You mention translating your dataset into "ten typologically diverse languages" (line 359). Which languages are included and what makes this selection typologically diverse? If it is the languages reported appendix E, then the selection does not seem so diverse, or at least skewed (German, Spanish, French, …). Would this be quantifiable? Otherwise, this claim is not justified. This question also applied to "five diverse languages" (line 478).**
>
> For the translations mentioned in L359, we strictly adhered to the MGSM benchmark setup (Shi et al., 2023) to ensure fair comparison with prior work. We used the exact same set of 10 languages defined in their work as "typologically diverse" (Bengali, German, Spanish, French, Japanese, Russian, Swahili, Telugu, Thai, Chinese). We agree that the term "diverse" is relative, but we adopted it to maintain consistency with standard literature terminology (Shi et al., 2023).
> For GeoFact-X (L478), our subset (English, Hindi, Japanese, Swahili, Thai) spans four distinct scripts (Latin, Devanagari, Kana/Kanji, Thai) and multiple language families. Notably, Swahili (Bantu family) provides structural diversity distinct from Indo-European languages despite using the Latin script.

---

> ### Author Response · Authors · 2025-11-21
> **Author Response (1/3)**
>
> **Re: Can you assume that Gemini 2.0 generates reasoning traces consistently for each of the five languages when you use it to create the GeoFact-X dataset (lines 269-270)? If so, is reasoning language consistency really a prominent issue in LLMs?**
>
> First, regarding dataset creation, Gemini 2.0 is not perfectly consistent. During generation, we observed multiple failures, particularly in lower-resource settings (e.g., mixing Swahili with English, or confusing Japanese and Thai scripts). This necessitated the strict filtering and human verification steps described in our paper to ensure GEOFACT-X quality.
>
> Second, and more critically, the "issue" we address is not the lack of capability in base models, but the catastrophic degradation of that capability during post-training. As shown in Section 6.2 (Table 2), while the base Qwen model starts with high consistency, applying standard SFT causes it to collapse (Language Accuracy drops $99.0 \rightarrow 31.0$). The central motivation of our work is that standard methods force a trade-off: you can have better math (SFT) or better language consistency (Base), but not both. M2A is significant precisely because it allows models to learn reasoning without suffering this training-induced multilingual collapse.
>
> **Re: "We use Qwen-2.5-7B-Instruct as the backbone for all experiments on mathematical and factual reasoning." (line 316-317): Why did you choose Qwen-2.5-7B-Instruct specifically? (Why) would you expect similar results for different models?**
>
> We selected Qwen-2.5-7B-Instruct as it was widely recognized as a state-of-the-art open-weight model in the 7B class, consistently outperforming peers (e.g., Llama-3.1, Mistral) on multilingual reasoning, mathematics, and coding benchmarks.
> To address the reviewer’s concern regarding generalizability, we conducted additional experiments using Command R7B (Cohere AI, 2025), another architecture specifically optimized for multilingual tasks and RAG. As shown in Table ZU1X-1 below, the results are consistent with our main Qwen findings. M2A achieves the highest performance in both Reasoning ($46.8$) and Answer ($30.9$) accuracy, outperforming SFT and standard GRPO. This confirms that our method is model-agnostic and effective across different multilingual architectures.
>
> **Table ZU1X-1. Performance on GeoFact-X using Command R7B Backbone**
>
> |                    | Reasoning        |                  || Answer           |||
> | ------------------ | ---------------- | ---------------- | ---------------- | ---------------- | ---------------- | ---------------- |
> |  | All              | Assoc.           | Non-Assoc.       | All              | Assoc.           | Non-Assoc.       |
> | Command R7B  | 44.2 | 48.5 | 43.1 | 25.0 | 34.1 | 22.7 |
> | GRPO      | 38.3 $\pm$ 0.4  |  42.4 $\pm$ 0.9  |  37.2 $\pm$ 0.4  |  23.4 $\pm$ 3.2  |  30.5 $\pm$ 4.2  |  21.6 $\pm$ 3.0 |
> | SFT    | 46.6 $\pm$ 0.0 | **51.3 $\pm$ 0.3**  |  45.4 $\pm$ 0.1  |  28.9 $\pm$ 0.3  |  38.2 $\pm$ 0.6  |  26.6 $\pm$ 0.2 |
> | SFT + GRPO | 44.1 $\pm$ 1.4  |  48.8 $\pm$ 1.9  |  42.9 $\pm$ 1.2  |  21.9 $\pm$ 6.8  |  29.0 $\pm$ 8.9  |  20.1 $\pm$ 6.3
> | M2A      | **46.8 $\pm$ 0.8**  |  50.1 $\pm$ 0.9  |  **46.0 $\pm$ 0.7**  | **30.9 $\pm$ 0.9** |  **38.5 $\pm$ 1.3**  |  **29.1 $\pm$ 0.8** |
>
> **Re: Why, in Tables 3 and 4, do you specifically evaluate Thai only?**
>
> We clarify that the label "M2A (Thai only)" refers specifically to the training configuration, where the alignment language $l$ was restricted to Thai (as detailed in L424-426), rather than sampling from all five GeoFact-X languages. However, the evaluation was performed across all five languages to assess cross-lingual transfer capabilities.
>
>
> **Re: Caption in Table 3**
> We apologize for the error in the Table 3 caption, which was a labeling oversight. As defined in L278-280, the "Reasoning Score" metric explicitly incorporates language accuracy as a component. We will correct the caption in the revision.
>
> **Re: Typos**
> Thank you for identifying these. We have corrected them in the updated manuscript.

---

### Official Review · Reviewer_yDoH · 2025-10-30

**Soundness:** 2
**Presentation:** 3
**Contribution:** 2
**Rating:** 6
**Confidence:** 3

**Summary:**

LLMs ability to reason in low resource languages is still lacking. Even if models accurately answer the question in the low resource languages, the reasoning traces are often in English. The paper introduces M2A, an approach to do multi-scale multilingual alignment along with language consistency rewards which trains the model to reason in the same language as the question. M2A uses SFT + GRPO on ground-truth reasoning traces. The paper also introduces GEOFACT-X, a geography based multilingual factual reasoning benchmark along with reasoning traces in five languages. The authors also introduce a new metric to measure language accuracy since existing multilingual benchmarks assess the final accuracy but overlook the language of the reasoning traces.

**Strengths:**

The paper addresses a critical issue of interpretability in low resource languages due to lack of language adherence in LLM reasoning traces. Models often reason in English instead of the target language of the question. The paper introduces a reward modeling approach that combines rewards for language consistency, context and reasoning alignment. The paper also introduces a high quality dataset GEOFACT-X, a geography based multilingual factual reasoning benchmark, in five languages:  English, Hindi, Japanese, Swahili, and Thai. The paper has ablations to show why each component of the reward modeling is important.

**Weaknesses:**

1. While M2A shows improvements on Math or language accuracy over SFT and GRPO individually, it doesn't seem to improve MGSM Math or Language accuracy scores over the baseline Qwen-2.5-Instruct model in Table 2.

2. The proposed M2A approach requires ground truth outputs as well as ground truth reasoning traces for Multilingual Context Alignment and Multilingual Reasoning-Step Alignment as described in Section 3. This is a major limitation especially for resource constrained languages. One of the main benefits of online RL recipes such as GRPO is that you don't need ground truth responses.

**Questions:**

1. The baselines and experiment setup is not very clear. What reward model does baseline GRPO recipe use? Does M2A use SFT + GRPO or does it only use GRPO with the reward modeling mentioned in Section 3?

2. Is there a SFT + GRPO (with sparse rewards) baseline for comparison that can be added to Table 2?

---

> ### Author Response · Authors · 2025-11-20
>
> We thank the reviewer for their thoughtful comments and the opportunity to improve our work. We have carefully considered all points.
>
> **Re: While M2A shows improvements on Math or language accuracy over SFT and GRPO individually, it doesn't seem to improve MGSM Math or Language accuracy scores over the baseline Qwen-2.5-Instruct model in Table 2.**
>
> Our goal is to identify a post-training method that improves mathematical reasoning without **compromising** multilingual capabilities. Table 2 demonstrates that standard methods fail to manage this trade-off on MGSM.
> While Supervised Fine-Tuning (SFT) on s1K-1.1 increases MGSM Math accuracy ($58.7 \rightarrow 66.7$), it causes severe degradation in language accuracy ($99.0 \rightarrow 31.0$) and joint accuracy ($58.1 \rightarrow 21.9$), indicating the model frequently responds in the incorrect language or format. Conversely, GRPO leaves performance virtually unchanged: Math and joint accuracy remain near baseline levels with a slight drop in language accuracy ($99.0 \rightarrow 95.9$), offering no meaningful improvement in either dimension.
>
> M2A occupies a distinct position: it yields a modest gain in MGSM Math accuracy over the base model ($58.7 \rightarrow 59.0$) while preserving high language accuracy ($99.0 \rightarrow 97.8$) and maintaining joint accuracy ($58.1 \rightarrow 58.1$). Crucially, it simultaneously achieves the best performance on all GSM8K metrics. Thus, M2A is the only post-training strategy in Table 2 that enhances reasoning while maintaining strong multilingual fidelity, avoiding both the multilingual collapse of SFT and the stagnation of GRPO.
>
> **Re: The proposed M2A approach requires ground truth outputs as well as ground truth reasoning traces for M2A as described in Section 3. This is a major limitation especially for resource constrained languages. One of the main benefits of online RL recipes such as GRPO is that you don't need ground truth responses.**
>
> We acknowledge that requiring supervision can be a constraint. However, strictly speaking, M2A does not require *human* ground truth reasoning traces. In our experiments, we utilize LLM-generated pseudo-labels for both factual and mathematical reasoning (see Muennighoff et al., 2025, and Section 4). This makes M2A scalable even for languages where human annotations are unavailable.
>
> Furthermore, if human-annotated ground truth is desired, M2A only requires it in a single source language; our alignment objectives then transfer this supervision to target languages via multilingual context and step alignment. Finally, M2A is complementary to online RL methods like GRPO: our method exploits dense token-level supervision (via pseudo-traces), whereas RL methods operate with sparser scalar rewards.
>
>
> **Re: What reward model does baseline GRPO recipe use?**
>
> Following the DeepSeek-R1 (Guo et al., 2025), our GRPO baseline employs a rule-based reward model comprising three components: format correctness, answer accuracy, and language consistency. Full implementation details are provided in the supplementary code.
>
>
> **Re: Does M2A use SFT + GRPO or does it only use GRPO with the reward modeling mentioned in Section 3?**
>
> M2A jointly optimizes both SFT and GRPO objectives concurrently.
>
>
> **Re: Is there a SFT + GRPO (with sparse rewards) baseline for comparison that can be added to Table 2?**
>
> We have conducted this experiment and included the results in the revised Table 2 (Table-yDoH-1). As shown below, we observe that a naive combination of SFT and GRPO (using sparse outcome rewards) leads to severe training instability and performance collapse (Reasoning: $47.6 \rightarrow 26.9$; Answer: $29.3 \rightarrow 10.7$). This suggests that sparse scalar rewards are insufficient to guide the model through the complex optimization landscape of multilingual reasoning, often destabilizing the alignment achieved by SFT. In contrast, M2A, which utilizes dense token-level supervision and a joint alignment objective, avoids this collapse and successfully improves upon the strong SFT baseline.
>
> **Table-yDoH-1. Comparison with SFT + GRPO Baseline (GeoFact-X)**
>
> |Translation Method | Reasoning| | | Answer| | |
> | ---- | ----- | --- | --- | ------ | ---- | ---- |
> |  | All| Assoc.| Non-Assoc.| All| Assoc. | Non-Assoc.|
> | Qwen-2.5-Instruct  | 30.4 | 38.5 | 28.3 |  26.2 | 33.7 | 24.3|
> | GRPO  | 45.4 $\pm$ 0.2 | 48.1 $\pm$ 0.1 | 44.8 $\pm$ 0.3 | 32.1 $\pm$ 0.3 | 37.6 $\pm$ 0.2 | **30.7 $\pm$  0.3** |
> | SFT | 47.6 $\pm$ 0.1 | 50.7 $\pm$ 0.5 | 46.9 $\pm$ 0.1 | 29.3 $\pm$ 0.2 | 37.1 $\pm$ 1.0 | 27.3 $\pm$ 0.3 |
> | SFT + GRPO | 26.9 $\pm$0.9 | 29.2 $\pm$ 0.9 | 26.4 $\pm$ 0.9 | 10.7 $\pm$ 0.9 | 14.7  $\pm$  1.7 | 9.7  $\pm$  0.7 |
> | M2A|  48.5 $\pm$ 0.4 | 52.6 $\pm$ 0.5 | 47.5 $\pm$ 0.3 | 32.0 $\pm$ 0.6 | **41.3 $\pm$ 1.0** | 29.7 $\pm$ 0.5
> |M2A (Thai only) | **49.8 $\pm$ 0.3** | **53.4 $\pm$ 0.2** | **48.8 $\pm$ 0.5** | **32.2 $\pm$ 0.4** | 39.9 $\pm$ 0.4 | 30.2 $\pm$ 0.3

---

### Comment · Area_Chair_qPhc · 2025-11-22
**Action Needed: Review Rebuttal and Update Evaluation**

Dear Reviewers,

Thank you, as always, for your valuable contributions and efforts. The authors have now submitted their rebuttal. Please take a moment to review it and provide any necessary follow-up actions, such as additional questions, clarification requests, or updates to your review.

Since the initial ratings ranged from 2 to 6, I kindly ask you to pay close attention to the perspectives of the other reviewers when preparing your final response.

Thank you again for your support.

---

### Meta-Review · Area_Chair_2q38 · 2026-01-07

**Summary:**

This paper addresses language-consistent reasoning in LLMs through M2A (a post-training method) and GeoFact-X.

After rebuttal, the reviewers still hold several concerns. For example, the novelty and generalizability of the overall methodological framework have not been sufficiently demonstrated. The evaluation is limited to qwen backbone (which has been extended with Command R7B during discussion) and limited model size. The rigorous of the LLM-based evaluation using references needs to be validated, and the experimental settings for comparison against the base SFT baseline require further clarification.

**Reviewer Concerns:**

see metareview

**Reviewer Scores:**

see metareview

---

### Decision · Program_Chairs · 2026-01-26

Reject